# The cAMP effector PKA mediates Moody GPCR signaling in *Drosophila* blood–brain barrier formation and maturation

Xiaoling Li[1,2,3]*, Richard Fetter[4], Tina Schwabe[2†], Christophe Jung[2], Liren Liu[5], Hermann Steller[3]*, Ulrike Gaul[2,3]

[1]Tianjin Cancer Hospital Airport Hospital, Tianjin Medical University Cancer Institute & Hospital, Tianjin, China; [2]Department of Biochemistry, Gene Center, Center of Integrated Protein Science (CIPSM), University of Munich, Munich, Germany; [3]Rockefeller University, New York, United States; [4]Janelia Farm Research Campus, Howard Hughes Medical Institute, Ashburn, United States; [5]Department of Gastrointestinal Cancer Biology, Tianjin Medical University Cancer Institute & Hospital; National Clinical Research Center for Cancer; Key Laboratory of Cancer Prevention and Therapy; Tianjin's Clinical Research Center for Cancer, Tianjin, China

**\*For correspondence:**
lixiaoling@tmu.edu.cn (XL);
steller@rockefeller.edu (HS)

**Present address:** [†]Alector Pharmaceuticals LLC, San Francisco, United States

**Competing interest:** The authors declare that no competing interests exist.

**Abstract** The blood–brain barrier (BBB) of *Drosophila* comprises a thin epithelial layer of subperineural glia (SPG), which ensheath the nerve cord and insulate it against the potassium-rich hemolymph by forming intercellular septate junctions (SJs). Previously, we identified a novel Gi/Go protein-coupled receptor (GPCR), Moody, as a key factor in BBB formation at the embryonic stage. However, the molecular and cellular mechanisms of Moody signaling in BBB formation and maturation remain unclear. Here, we identify cAMP-dependent protein kinase A (PKA) as a crucial antagonistic Moody effector that is required for the formation, as well as for the continued SPG growth and BBB maintenance in the larva and adult stage. We show that PKA is enriched at the basal side of the SPG cell and that this polarized activity of the Moody/PKA pathway finely tunes the enormous cell growth and BBB integrity. Moody/PKA signaling precisely regulates the actomyosin contractility, vesicle trafficking, and the proper SJ organization in a highly coordinated spatiotemporal manner. These effects are mediated in part by PKA's molecular targets MLCK and Rho1. Moreover, 3D reconstruction of SJ ultrastructure demonstrates that the continuity of individual SJ segments, and not their total length, is crucial for generating a proper paracellular seal. Based on these findings, we propose that polarized Moody/PKA signaling plays a central role in controlling the cell growth and maintaining BBB integrity during the continuous morphogenesis of the SPG secondary epithelium, which is critical to maintain tissue size and brain homeostasis during organogenesis.

## Introduction

The blood–brain barrier (BBB) is a complex physical barrier between the nervous system and the peripheral circulatory system that regulates *central nervous system (CNS)* homeostasis to ensure proper neuronal function. The *Drosophila* BBB is established by a thin epithelium of subperineural glia (SPG), which ensheath and insulate the nervous system against the potassium-rich hemolymph by forming intercellular septate junctions (SJs) (*Bainton et al., 2005*; *Carlson et al., 2000*; *Edwards et al., 1993*). The SPG epithelium is formed as a result of a mesenchymal–epithelial transition (MET), similar to other secondary epithelia such as heart and midgut. SPG cells only increase in number in embryogenesis but not in larval development, and rather increase their size by polyploidization (*Unhavaithaya and Orr-Weaver, 2012*). Polyploidy in SPG is necessary to coordinate cell growth and

BBB integrity either by Notch signaling or miR-285–Yki/Mask signaling during CNS development at the larval stage (*Li et al., 2017*; *Unhavaithaya and Orr-Weaver, 2012*; *Von Stetina et al., 2018*). SPG cells lack the apical markers present in primary epithelia (Crumbs, Bazooka), they have no contiguous zonula adherens and therefore rely on their SJ belt for epithelial cohesion, preventing paracellular diffusion and sealing the BBB (*Schwabe et al., 2005*; *Stork et al., 2008*; *Tepass et al., 2001*).

SJs are the crucial barrier junctions in invertebrates and functionally equivalent to vertebrate tight junctions (TJs); both junctions share claudins as key components (*Izumi and Furuse, 2014*). Structurally and molecularly, SJs are homologous to the vertebrate paranodal junction that forms between axons and myelinating glial cells adjacent to the node of Ranvier (for review, see *Banerjee et al., 2006*; *Salzer, 2003*; *Salzer, 2015*; *Salzer et al., 2008*). They consist of a core mutual interdependence protein complex, including transmembrane and cytoplasmic proteins, such as Neurexin-IV (Nrx-IV), Neuroglian (Nrg), the Na/K-ATPase (ATPα and Nrv2), the claudin Megatrachea (Mega), Sinous, Coracle (Cora), and the tetraspan Pasiflora protein family (*Oshima and Fehon, 2011*). In addition to the above-listed proteins, several GPI-anchored proteins, including Ly6-domain proteins Boudin, Crooked, Crimpled, and Coiled, Lachesin, Contactin, the tetraspan Pasiflora protein family, and Undicht, which are all found to be required for the SJ complex formation and proper membrane trafficking (*Deligiannaki et al., 2015*; *Faivre-Sarrailh et al., 2004*; *Hijazi et al., 2011*; *Hijazi et al., 2009*; *Llimargas et al., 2004*; *Petri et al., 2019*; *Tempesta et al., 2017*). The intracellular signaling pathways that control the assembly and maintenance of SJs are just beginning to be elucidated.

We have previously identified a novel GPCR signaling pathway that is required for the proper organization of SJ belts between neighboring SPG at the embryonic stage, consisting of the receptor Moody, two hetero-trimeric G proteins (Gαiβγ, Gαoβγ), and the RGS protein Loco. Both gain and loss of Moody signaling lead to non-synchronized growth of SPG cells, resulting in disorganized cell contacts and shortened SJs and, therefore, a leaky BBB (*Schwabe et al., 2005*; *Schwabe et al., 2017*). The phenotype of Moody is weaker than that of downstream pathway components including Loco and Gβ13F, suggesting that additional receptors provide input into the trimeric G protein signaling pathway. Gγ1 signaling has been shown to regulate the proper localization of SJ proteins in the embryonic heart (*Yi et al., 2008*). Despite its critical role in BBB formation, the underlying mechanisms connecting G protein signaling to continued SPG cell growth and the proper SJ organization during the development and maturation of BBB are still poorly understood.

One of the principal trimeric G protein effectors is adenylate cyclase (AC). AC is inhibited by the G proteins Gαi/Gαo and Gβγ, leading to decreased levels of the second messenger cAMP. The prime effector of cAMP, in turn, is cAMP-dependent protein kinase A (PKA), a serine/threonine kinase. PKA is inactive as a tetrameric holoenzyme, which consists of two identical catalytic and two regulatory subunits. Binding of cAMP to the regulatory units releases and activates the catalytic subunits (*Taylor et al., 1990*). PKA transmits the signal to downstream effectors by phosphorylating multiple substrates that participate in many different processes, from signal transduction to regulation of cell shape and ion channel conductivity (*Shabb, 2001*). In *Drosophila*, PKA has been studied as a component of GPCR signaling in the Hedgehog pathway during development (*Li et al., 1995*; *Marks and Kalderon, 2011*), and in neurotransmitter receptor pathways during learning and memory (*Chen and Ganetzky, 2012*; *Guan et al., 2011*; *Li et al., 1996*; *Renger et al., 2000*). PKA also regulates microtubule organization and mRNA localization during oogenesis (*Lane and Kalderon, 1993*; *Lane and Kalderon, 1994*; *Lane and Kalderon, 1995*). In vertebrates, cAMP/PKA signaling is known to play a central role within different subcellular regions, including the regulation of actomyosin contractility and localized cell protrusion in directional cell migration (*Howe, 2004*; *Lim et al., 2008*; *Tkachenko et al., 2011*); intracellular membrane trafficking (exocytosis, endocytosis, and transcytosis) in relation to the dynamics of epithelial surface domains in developmental processes and organ function (*Wojtal et al., 2008*). Moreover, cAMP/PKA signaling regulates endothelial TJ with diverse actions and unclear mechanisms in different endothelial cells models (*Cong and Kong, 2020*).

Here, we report results from a comprehensive in vivo analysis of the molecular and cellular mechanisms of Moody signaling in the SPG. We show that PKA is a key downstream effector responsible for the salient phenotypic outcomes, and that it acts by modulating actomyosin contractility via MLCK and Rho1. The strong phenotypic effects of PKA gain- and loss of function permit a detailed dissection of the organization of cell–cell contacts as driven by Moody/PKA signaling and allow us to track its role in the continued growth of the SPG during larval stages. We observe asymmetric and opposing

subcellular distributions of Moody and PKA, providing novel insight into the establishment of apical–basal polarity in the SPG as a secondary epithelium, as well as its morphogenetic function. We present a 3D reconstruction of SJ ultrastructure using serial section transmission electron microscopy (ssTEM) under different PKA activity levels. This new analysis reveals a strict coupling of total cell contact and SJ areas, but also suggests that it is the continuity of individual SJ segments and not total SJ width that is essential for normal BBB insulation. Altogether, our data reveal a previously unrecognized role of GPCR/PKA in maintaining enormous SPG cell growth and its sealing capability by regulating acto-myosin contractility and the proper SJ organization in BBB formation and maturation, which touches the fundamental aspects of remodeling cytoskeletal network spatiotemporally – a common process but with different mechanisms in morphogenesis.

## Results

### PKA is required for Moody-regulated BBB formation

To identify molecules that act downstream of Moody signaling in BBB formation, we examined genes known to be involved in GPCR signaling, such as PkaC1, PI3K, PTEN, PLC, and Rap1. We tested BBB permeability in genomic mutants or transgenic RNAi knockdowns of these GPCR effectors by injecting a fluorescent dye into the body cavity and determining its penetration into the CNS using confocal imaging. We found that zygotic mutants of the PKA catalytic subunit PkaC1 (originally named DC0 in *Drosophila*), namely, the two null alleles *PkaC1^B3^* and *PkaC1^H2^* as well as the hypomorphic allele *PkaC1^A13^* (*Kalderon and Rubin, 1988*), show severe CNS insulation defects (*Figure 1A and B*), similar in strength to zygotic mutants of the negative regulator *loco*. By contrast, the removal of the other candidates had no effect (data not shown). *PkaC1* has both maternal and zygotic components, and its maternal contribution perdures until late embryogenesis (*Lane and Kalderon, 1993*). The BBB defect we observe could explain the morphologically inconspicuous embryonic lethality of *PkaC1* zygotic null mutants (*Lane and Kalderon, 1993*). To rule out the possibility that the observed BBB defects are caused by glial cell fate or migration defects, we examined the presence and position of SPG using an antibody against the pan-glial, nuclear protein Reversed polarity (Repo) (*Halter et al., 1995*). In *PkaC1* zygotic mutants, the full set of SPG is present on the surface of the nerve cord, although the position of the nuclei is more variable than in WT (*Figure 1C*), an effect that is also observed in known mutants of the Moody signaling pathway (*Granderath et al., 1999*; *Schwabe et al., 2005*).

Since SJs are the principal structure providing BBB insulation and are disrupted in Moody pathway mutants (*Schwabe et al., 2005*; *Schwabe et al., 2017*), we sought to characterize the SJ morphology in PkaC1 mutants. We performed ultrastructural analysis of SJs in late embryos (*after egg lay (AEL) 22–23 hr*) by TEM using high-pressure freezing fixation. In WT, the SJs are extended, well-organized structures that retain orientation in the same plane over long distances (*Figure 1D*). In contrast, in *PkaC1^H2^* zygotic mutants, the overall organization of SJs appears perturbed, and their length, as measured in random single sections, is significantly shorter than in WT (0.31 ± 0.03 µm vs. 0.57 ± 0.07 µm, p=0.000457; *Figure 1D and E*); very similar phenotypic defects are observed in *moody* and *loco* zygotic mutants (*Schwabe et al., 2005*).

To explore the role of PkaC1 during development of the BBB, we performed time-lapse recordings of SPG epithelium formation. The SPG arise in the ventrolateral neuroectoderm and migrate to the surface of the developing nerve cord (*Ito et al., 1995*), where they spread until they reach their neighbors and form intercellular SJs (*Schwabe et al., 2005*; *Schwabe et al., 2017*). To monitor the changes in SPG morphology during the closure process, we expressed the membrane marker *GapGFP* and the actin marker *MoesinGFP* using the pan-glial driver *repoGAL4* (*Schwabe et al., 2017*; *Figure 1*, *Figure 1—video 1*, *Figure 1—video 2*). In WT embryos, SPG are relatively uniform in cell size and shape, and grow to form cell–cell contacts in a highly synchronized manner. By 15.5 hr of development, the glial sheet is closed (*Figure 1F*). By contrast, SPG in *PkaC1^H2^* zygotic mutants show increased variability in size and shape, and their spreading and contact formation is less well coordinated. This results in patchy cell–cell contacts with gaps of variable sizes (*Figure 1F*). Moreover, the complete closure of the SPG epithelium is delayed compared to WT (*Figure 1F*). Again, the defects observed in PKA loss of function are similar to those in Moody pathway mutants (*Schwabe et al., 2017*).

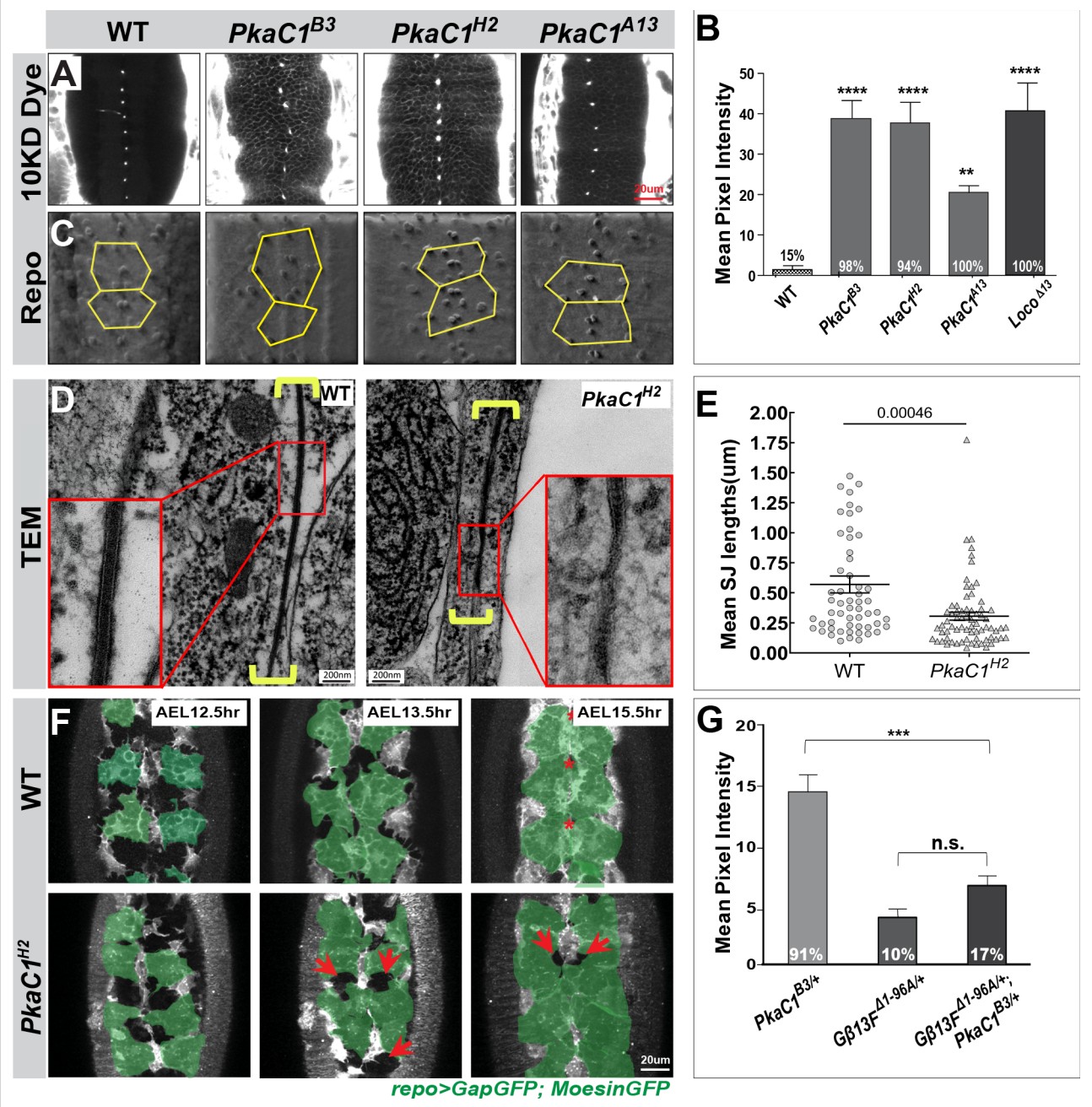

**Figure 1.** Protein kinase A (PKA) is required for blood–brain barrier (BBB) formation and acts in the Moody signaling pathway. (**A**) Single confocal sections of dye-injected embryos of WT and PKA zygotic mutants. (**B**) Quantification of the dye penetration assay. Columns represent the intensity of dye penetration into the nerve cord as measured by the mean pixel intensity (see Experimental procedures), ± SEM, n = 32, 31, 41, 38, 16 in WT, $PkaC1^{B3}$, $PkaC1^{H2}$, $PkaC1^{A13}$, $Loco^{\Delta13}$ embryos, respectively. $Loco^{\Delta13}$ zygotic mutants serve as positive controls. (**C**) Repo staining revealing the number and positions of subperineural glia (SPG) nuclei in WT and PKA zygotic mutants using an illuminated projection to highlight the ventral surface of the nerve cord. (**D**) Transmission electron micrographs of the interface of neighboring SPG in late WT and $PkaC1^{H2}$ zygotic mutant embryos. Yellow brackets delineate the septate junction (SJ) ultrastructure; high magnifications are shown in red boxes. (**E**) Quantification of SJ length in WT and $PkaC1^{H2}$ mutants (see Experimental procedures). Columns represent mean SJ length as measured in random nerve cord sections, ± SEM, n = 56 and n = 70 in WT and $PkaC1^{H2}$ mutants, respectively. (**F**) Time-lapse recording of BBB closure in embryos of WT and PKA zygotic mutants. 6 μm confocal stacks are shown; in each image, 4–6 ventral SPG are highlighted (green); midline channels (stars) and retarded growth (arrows) are marked. (**G**) Dominant genetic interactions between $PkaC1^{B3}$ and $G\beta13F^{\Delta1-96A}$ as quantified by dye penetration in the embryo. Columns represent the intensity of dye penetration as measured by the mean pixel intensity, ± SEM, n = 34, n = 48, and n = 71 in $PkaC1^{B3/+}$, $G\beta13F^{\Delta1-96A/+}$, and $G\beta13F^{\Delta1-96A/+};PkaC1^{B3/+}$ mutants, respectively. In (**B**) and (**G**), the percentage of embryos showing the dye penetration is indicated at the bottom of each column. Brackets and asterisks in (**B**), (**E**), and (**G**) indicate statistical significance levels as assessed by ordinary one-way ANOVA with Dunnett's multiple comparisons test in (**B**) and (**G**) or the two-tailed

*Figure 1 continued on next page*

*Figure 1 continued*

Student's t-test in (**E**), n.s., p>0.05; *p<0.05; **p<0.01; ***p<0.001.

The online version of this article includes the following video and figure supplement(s) for figure 1:

**Figure 1—video 1.** Subperineural glia (SPG) epithelium formation in WT embryos.

https://elifesciences.org/articles/68275/figures#fig1video1

**Figure 1—video 2.** Subperineural glia (SPG) epithelium formation in protein kinase A (PKA) zygotic null mutant embryos.

https://elifesciences.org/articles/68275/figures#fig1video2

**Figure supplement 1.** Genetic interactions between protein kinase A and different Moody pathway components.

Our results show that *PkaC1* is required for BBB integrity, proper SJ organization, and SPG epithelium formation, in all cases closely mimicking the phenotypes observed for known Moody signaling components. Given these similarities, we sought to determine whether PKA participates in the Moody pathway by performing dominant genetic interaction experiments. Notably, we found that embryos heterozygous for *PkaC1* null alleles, which are known to have ~50% of wildtype PkaC1 activity, show mild BBB permeability defects (*Figure 1G*). Therefore, we used *PkaC1^B3* heterozygous mutants as a sensitized genetic background and removed one genomic copy of different Moody pathway components, including Moody, Loco, Gαo, Gαi, and Gβ13F (*Schwabe et al., 2005*), to determine whether any synergistic or antagonistic interactions are observed. We found that the dye penetration defects of *PkaC1* heterozygous mutants are significantly reduced by removing one genomic copy of *Gβ13F* or *loco* (*Figure 1G* and *Figure 1—figure supplement 1*, p=0.0022 or p=0.0115); removal of one genomic copy of *Gβ13F* or *loco* on their own has no effect. These genetic interactions indicate that PkaC1 is indeed part of the Moody signaling pathway. Removal of single copies of other pathway components showed either a mild, non-significant or no effect in a *PkaC1^B3* background, suggesting that they are less dosage-sensitive (*Figure 1—figure supplement 1*).

## PKA is required for BBB continued growth in larvae and BBB maintenance in adults

For a more detailed analysis of PkaC1 function in BBB regulation, we turned to the SPG epithelium in third instar larvae. During the larval stage, no additional SPG cells are generated, instead the existing SPG cells grow enormously in size to maintain integrity of the BBB (*Li et al., 2017*; *Unhavaithaya and Orr-Weaver, 2012*). By third instar, they have roughly doubled in size and are accessible via dissection of the CNS, which greatly facilitates the microscopic analysis. PKA activity in larvae can be manipulated specifically using the SPG-specific driver *moodyGAL4* (*Bainton et al., 2005*; *Schwabe et al., 2005*), which becomes active only after epithelial closure and BBB sealing are completed in stage 17 embryos. PKA can be reduced by expression of transgenic RNAi targeting the PKA catalytic subunit C1 (*moody>PkaC1* RNAi). On the other hand, PKA can be elevated by expression of a mouse constitutively active PKA catalytic subunit (*moody>mPkaC1*; *Zhou et al., 2006*). We first examined whether normal Moody/PKA activity is required for BBB integrity during larval stages. To address this question, we developed a dye penetration assay to measure BBB permeability in cephalic complexes of third instar larval. This assay is similar to the one we performed in the late embryo, but with some important modifications (for details, see Experimental procedures). Interestingly, both elevated and reduced activity of Moody (*moody>LocoRNAi* and *moody>moodyRNAi*) and PKA (*moody>mPKAC1** and *moody>PKAC1* RNAi) in SPG resulted in severe BBB insulation defects (*Figure 2A and D*). This strongly suggests that Moody/PKA signaling plays a crucial role in the continued growth of the BBB during larval stages. These effects were not merely carried over from the embryo since under *moody* driver caused only mild dye penetration defects in embryos (*Figure 2—figure supplement 1*). Given that Moody activity has been implicated in the maintenance of the BBB in the adult (*Bainton et al., 2005*), we also sought to knockdown PKA specifically in the adult SPG (*tubGal80ts, moody>PkaC1* RNAi) and measure the resulting effects (see Experimental procedures). We observed the dye significantly penetrated the blood–eye barrier under reduced PKA expression compared to that in WT (1.00 ± 0.34 vs. 6.38 ± 0.37, p<0.000001; *Figure 2F and G*), indicating that PKA is indeed also required for BBB integrity function in the adult.

In order to better understand the cause of BBB permeability under conditions where Moody/PKA is changed, we examined SJ morphology in larvae. Most core SJ components show interdependence

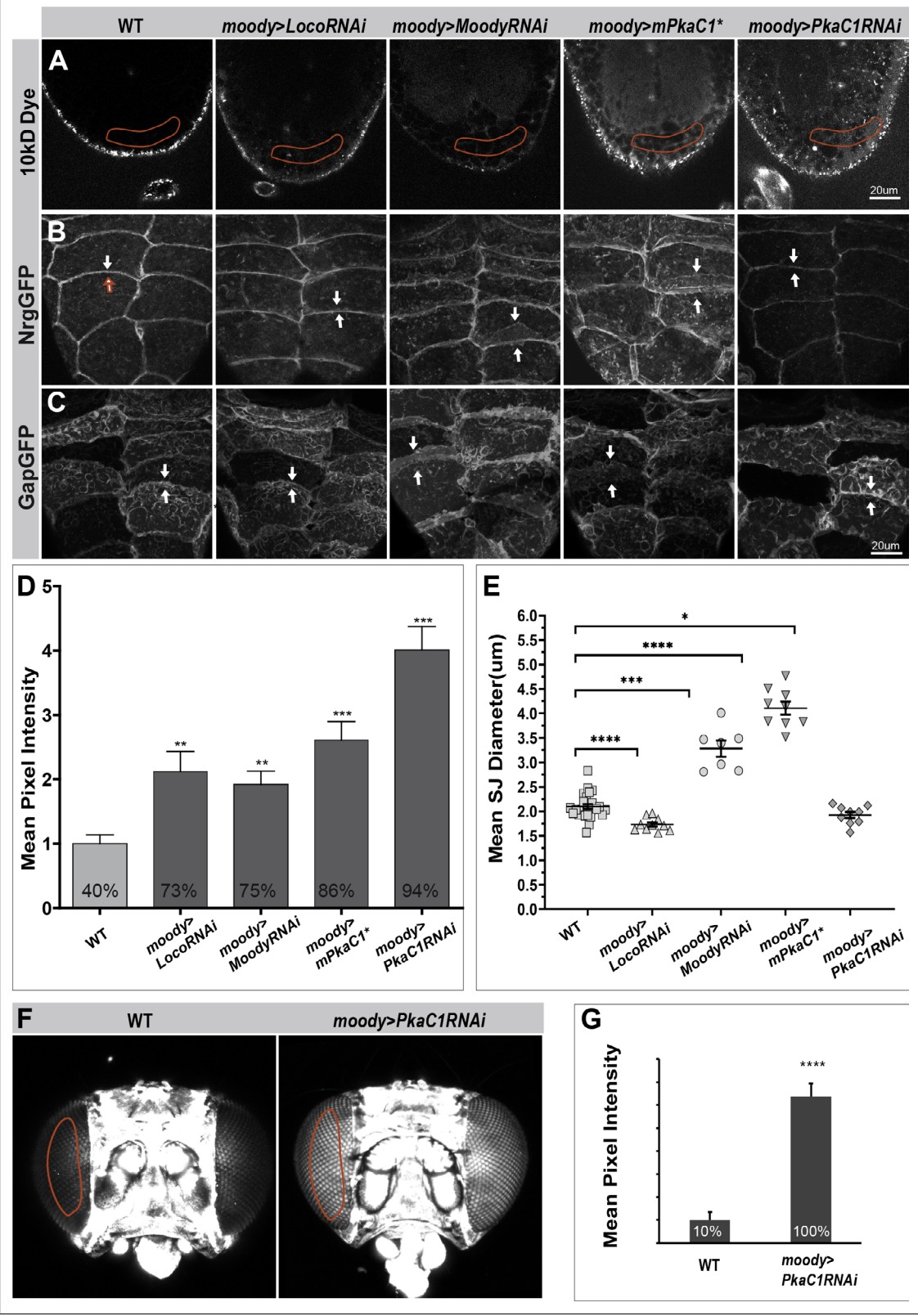

**Figure 2.** Moody/protein kinase A (PKA) signaling is required for blood–brain barrier (BBB) growth in the larva and for BBB maintenance in the adult. (**A**) Single confocal sections of dye-injected third instar larval nerve cords under different Moody/PKA activity levels. (**B, C**) Morphology of subperineural glia (SPG) septate junction (SJ) belts and membrane overlap at different Moody/PKA activity levels, as visualized by SJ markers NrgGFP (**B**), and the membrane marker GapGFP (**C**). (**D**) Quantification of the dye penetration assay. Columns represent intensity of dye penetration as measured by mean

*Figure 2 continued on next page*

*Figure 2 continued*

pixel intensity (see Experimental procedures), ± SEM, n = 44–88. The percentage of larva showing dye penetration is indicated at the bottom of each column. (**E**) Quantification of the diameter of SJ belts under different GPCR/PKA activity levels, using the SJ marker NrgGFP.± SEM, n = 7–28. (**F**) Dye penetration in adult flies as shown in z-projections of dye-injected adult heads. (**G**) Quantification of dye penetration in adult eye. Columns represent intensity of dye penetration as measured by mean pixel intensity in each adult eye (see Experimental procedures), ± SEM, n = 30 and 18. Asterisks in (**D**), (**E**), and (**G**) indicate significance levels of comparisons based on Welch's ANOVA with Dunnett's T3 multiple comparisons test (**D**) and (**E**) or the two-tailed unpaired t-test (**G**), n.s., p>0.05; *p<0.05; **p<0.01, ***p<0.001, ****p<0.0001.

The online version of this article includes the following figure supplement(s) for figure 2:

**Figure supplement 1.** Dye penetration defects for RNAi vs. genomic mutants in the embryo.

**Figure supplement 2.** Morphology of subperineural glia (SPG) septate junction (SJ) belts at different protein kinase A (PKA) activity levels, as visualized by SJ markers LacGFP (**A**), Mega (**B**), and Nrx-IVGFP (**C**).

for correct localization and barrier function, with removal of one component sufficient to abolish SJ function (*Behr et al., 2003*; *Genova and Fehon, 2003*; *Hijazi et al., 2011*; *Oshima and Fehon, 2011*; *Wu et al., 2004*). We therefore asked whether PKA activity levels affect the distribution of different SJ components. Using both live imaging (*NrgGFP, LacGFP, NrxIVGFP*) and immunohistochemistry (Mega), we found that the circumferential SJ belts and outlines of SPG were marked nicely in WT (*Figure 2B* and *Figure 2—figure supplement 2*). Strikingly, upon either reduction of Moody activity or elevated PKA activity, the SJ belt staining became much broader and more diffuse than in WT (*Figure 2B*). This suggests extensive plasma membrane overlap between neighboring SPG cells. To confirm this idea, we introduced the membrane marker *gapGFP*, and indeed observed increased membrane overlap compared to WT (*Figure 2C*). Conversely, both elevated Moody activity and reduced PKA activity resulted in thinner SJ belts and reduced membrane contacting area (*Figure 2B and C*). To quantify these changes, we measured the mean width of the SJ belts under different PKA activity levels (*Figure 2E*; Experimental procedures). The mean width of SJ belts increased with elevated PKA activity/reduced Moody activity and decreased under inverse conditions compared to WT (*Figure 2E*). These data demonstrate that Moody and PKA are required for the continued growth of the BBB and the proper organization of SJs during larval stages. Unlike the barrier defect, these morphological data reveal a monotonic relationship between PKA activity, membrane overlap, and the amount of SJ components in the area of cell contact. The fact that the cellular defects of reduced Moody activity match those of elevated PKA activity, and vice versa, provides further evidence that PKA acts as an antagonistic effector of Moody signaling.

## PKA regulates the cytoskeleton and vesicle traffic in SPG

We had previously reported that the Moody pathway regulates the organization of cortical actin and thus the cell shape of SPG during late embryogenesis (*Schwabe et al., 2005*; *Schwabe et al., 2017*). Moreover, we proposed, based on the developmental timeline, that this in turn affects the positioning of SJ material along the lateral membrane. Given that the most striking phenotype caused by altered PKA activity is the extent of membrane overlap, we sought to further explore if PKA functions by regulating the cytoskeleton in SPG.

For this purpose, we examined the intracellular distribution of the actin cytoskeleton in the SPG at different PKA levels. As live markers, we used *GFPactin*, which labels the entire actin cytoskeleton, *RFPmoesin* (*Schwabe et al., 2005*), that preferentially labels the cortical actin, the presumptive general MT marker *TauGFP* (*Jarecki et al., 1999*), the plus-end marker *EB1GFP* (*Rogers et al., 2004*), and the minus-end marker *NodGFP* (*Clark et al., 1997*; *Cui et al., 2005*; *Figure 3A–D*, and data not shown). In response to changes in PKA activity, all markers showed altered distributions similar to those observed with SJ markers. Specifically, elevated PKA activity caused all markers to become enriched at the cell cortex area, consistent with the broader membrane contacting area between neighboring SPG (*Figure 3A–D*, middle column). Conversely, upon reducing PKA activity, all markers were reduced or depleted from the cell cortex, consistent with reduced contact area between neighboring SPG (*Figure 3A–D*, right column). Thus, PKA signaling profoundly reorganizes the actin and MT cytoskeleton, which may affect the membrane contacting area between neighboring SPG.

Since PKA has been shown to affect vesicle trafficking in epithelial cells and neurons (*Renger et al., 2000*; *Vasin et al., 2014*; *Wojtal et al., 2008*; *Zhang et al., 2007*), we investigated if PKA signaling has a similar role during continued SPG cell growth. We introduced two live markers, *Rab4RFP*, which

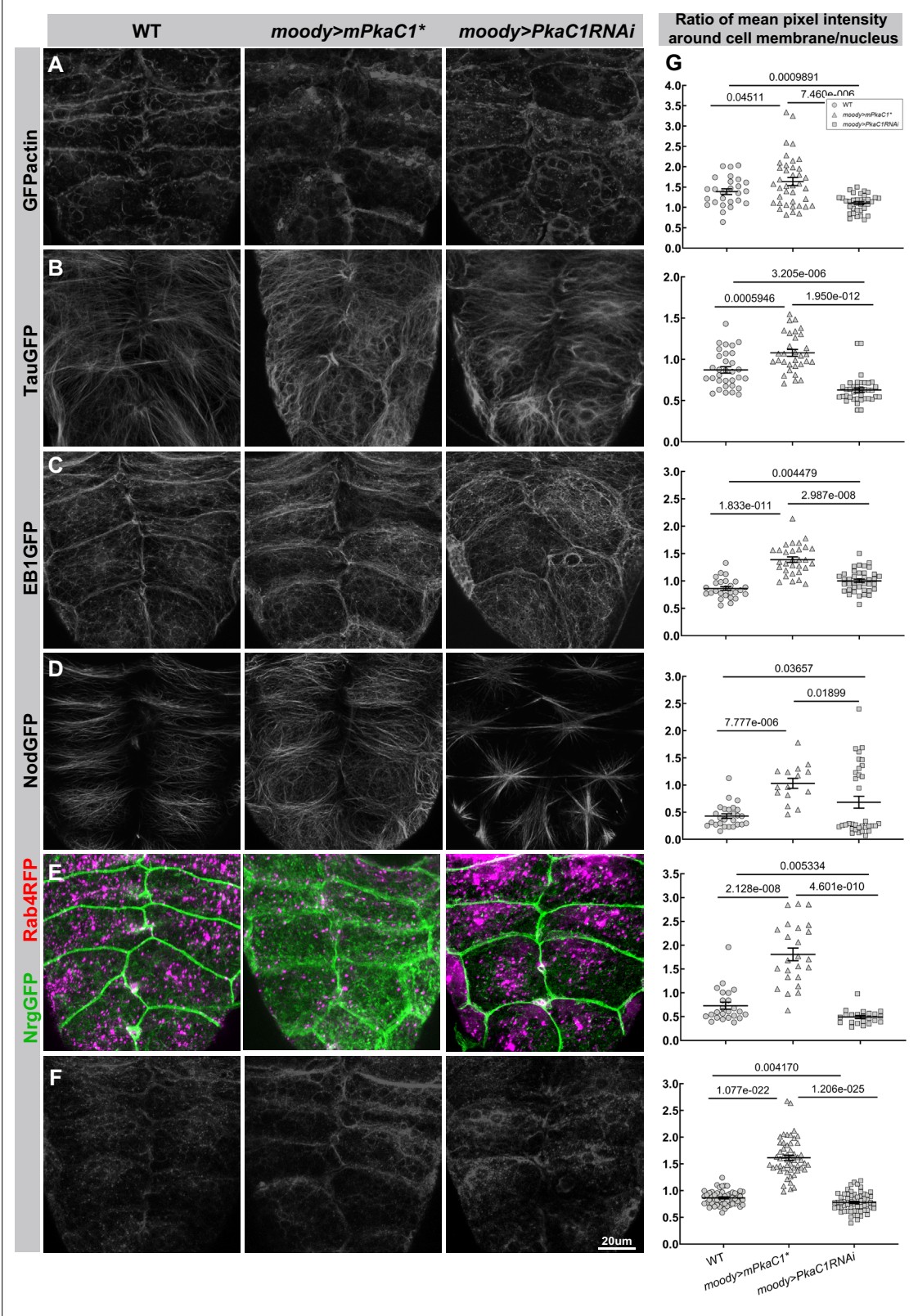

**Figure 3.** Protein kinase A (PKA) regulates the cytoskeleton and vesicle distribution in subperineural glia (SPG). Under different PKA activity levels, the actin cytoskeleton is visualized by GFPactin (**A**), the microtubule cytoskeleton by the general MT marker TauGFP (**B**), the plus-end marker EB1GFP (**C**), and the minus-end marker NodGFP (**D**), the cellular distribution of vesicles by the early endosome markers Rab4RFP (**E**) and Rab11GFP (**F**) with the septate junction (SJ) marker NrgGFP labeling the cell periphery of the SPG (**E**). (**G**) Quantification of the distribution of all markers around membrane

*Figure 3 continued on next page*

*Figure 3 continued*

area and the nucleus area under different PKA activity levels, including GFPactin, TauGFP, EB1GFP, NodGFP, Rab4RFP, and Rab11GFP. Columns represent the ratio of mean pixel intensity around cell membrane/nucleus in random SPG cells, ± SEM, n = 22–54. Asterisks indicate significance levels of comparisons based on Brown–Forsythe and Welch's ANOVA with multiple comparisons test, p value is label in each comparison group. A strong cortical actin rim around cell periphery is visible in WT (**A**). Compared with WT, the cortical actin rim is wider and stronger upon increased PKA activity and becomes weaker and less regular with reduced PKA activity. In WT (**B**–**D**), TauGFP-labeled microtubule fibers extend from MTOC to cell periphery throughout the cytoplasm (**B**); EB1GFP is enriched at the cell cortex but visible throughout the cytoplasm (**C**); NodGFP is enriched on fibers around the nucleus, but not around the cell cortex (**D**). Upon increased PKA activity, EB1GFP shows broader and more diffuse localization at the cell cortex, TauGFP and NodGFP become enriched at the cell periphery and show a web-like structure throughout the cytoplasm. Upon reduced PKA activity, TauGFP reveals disorganized and wavy microtubule organization; EB1 localization at the cell cortex is reduced; NodGFP accumulates around the MTOC in a striking star-shaped fashion. Rab4RFP- and Rab11GFP-labeled endosomes are differentially enriched in the cell periphery under PKA overactivity and surrounding the nucleus under reduced PKA activity when compared to its broader pan-cytoplasmic distribution in WT (**E**, **F**).

labels all the early endosomes (*Figure 3E*), and *Rab11GFP* (*Artiushin et al., 2018*), which labels both early and recycling endosomes (*Figure 3F*). We observed significant changes in the cellular distribution of vesicle populations. Specifically, Rab4- and Rab11-labeled endosomes were differentially enriched in the cell periphery when PKA activity is increased and surrounded the nucleus when PKA was reduced, as compared to their broader cytoplasmic distribution profile in WT (*Figure 3E and F*). Therefore, our results with these different cytoskeletal and vesicle markers suggest that the major function of PKA is regulating the cell contact areas between neighboring SPGs: high levels give rise to broad membrane contacts, and low levels of PKA activity cause narrow membrane contacts, which then affects SJ organization and BBB function.

## The continuity of SJ belt is essential for BBB function as revealed by ssTEM

While PKA gain- and loss of function show opposite morphologies of membrane overlap and SJ belt by light microscopy, they both result in a compromised leaky BBB. To better understand this incongruence, we sought to analyze membrane morphology at a higher resolution. Due to the small size of SJs (20–30 nm), structural aspects can be analyzed conclusively only by electron microscopy. In the past, the acquisition and analysis of a complete series of TEM sections required an enormous effort; as a consequence, studies of SJ structure have mostly been restricted to random sections (*Carlson et al., 2000*; *Hartenstein, 2011*; *Stork et al., 2008*; *Tepass and Hartenstein, 1994*). The problem has now become solvable, using digital image recording (*Suloway et al., 2005*) and specialized software (Fiji, TrakEM2)(*Cardona et al., 2012*; *Schindelin et al., 2012*) for both image acquisition and post-processing. Therefore, we performed serial section TEM, followed by computer-aided reconstruction of TEM stacks, to resolve the 3D ultrastructure of cell contacts and SJs under different PKA activity levels at third instar larva (*Figure 4A and B*). This is the first time that a contiguous SJ belt between neighboring SPG at nanometer resolution is presented.

In WT, the area of cell–cell contact is compact and well-defined, with a dense SJ belt covering ~30% of the cell contact area (*Figure 4A, B and F*). Upon elevated PKA activity, neighboring SPG show much deeper membrane overlap (*Figure 4A–E*). The areas of both cell contact and SJ coverage increase about twofold compared with WT (*Figure 4F*), confirming the observations from confocal microscopy (*Figure 3A–D*), but the SJ belt is discontinuous and appears patchy (*Figure 4B–E*). This suggests that it is the continuity of the belt, rather than the total area covered by SJs, that is essential for generating the intercellular sealing capacity. To examine this question directly, we measured SJ length in randomly selected sections. Compared with WT, the mean length of individual SJ segments (0.69 ± 0.08 μm vs. 2.16 ± 0.14 μm, p<0.0001) is indeed significantly decreased, while the mean total length of SJs (4.28 ± 0.43 μm vs. 2.16 ± 0.14 μm, p=0.000523) is significantly increased (*Figure 4G*).

Upon reducing PKA activity, the cell contacts and SJ area between neighboring SPGs were reduced, and the SJ belt became patchy as well (*Figure 4A–E*). In this case, both the mean total length of SJs (1.49 ± 0.08 μm vs. 2.16 ± 0.14 μm, p=0.000878) and the mean length of individual SJ segments (0.67 ± 0.14 μm vs. 2.16 ± 0.14 μm, p<0.0001) were significantly shorter than in WT (*Figure 4G*). Intriguingly, the ratio of total SJ area to cell contact area remains constant at about 30% under all PKA activity conditions, despite the variable interdigitations between contacting SPG (*Figure 4F*).

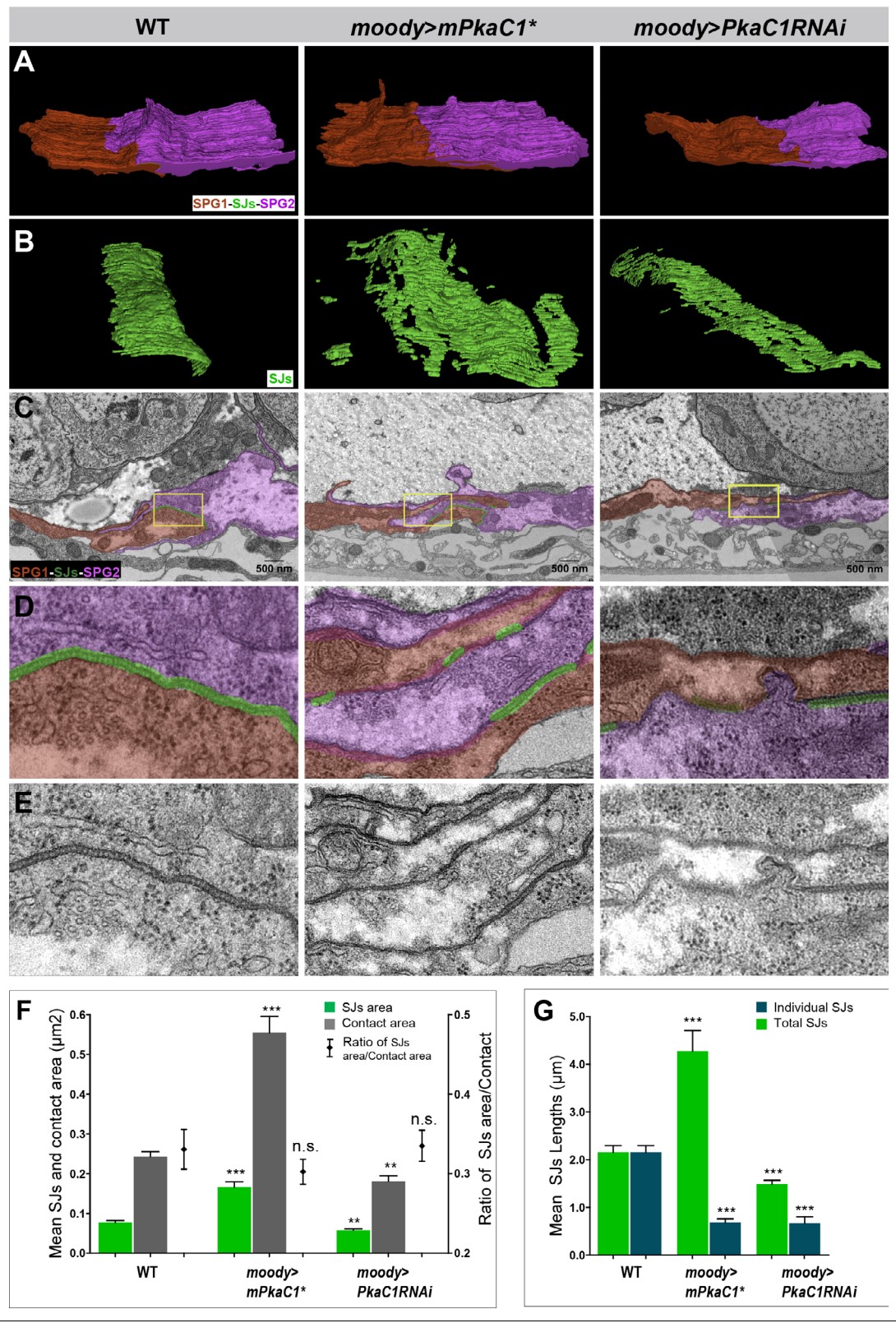

**Figure 4.** The continuity of the septate junction (SJ) belt is essential for blood–brain barrier (BBB) function as revealed by serial section transmission electron microscopy (ssTEM). (**A–E**) SJ ultrastructure at the interface of neighboring subperineural glia (SPG) in third instar larvae under different protein kinase A (PKA) activity levels. SPG1, its neighbor SPG2, and their shared SJs are colored or shaded in red, magenta, and green, respectively. (**A, B**) A 3D model of SJ ultrastructure generated by ssTEM. (**C**) Representative sections of SJs. (**D, E**) High-magnification views of boxed regions in (**C**) with and

*Figure 4 continued on next page*

*Figure 4 continued*

without shading. In WT, the area of contacting SPG is compact and well defined, and a dense SJ belt is formed between neighboring SPG. Under PKA overactivity (moody>mPkaC1*), neighboring SPG show much deeper interdigitations, and the SJs belt are discontinuous and appear patchy. Under PKA underactivity (*moody>PkaC1* RNAi), the cell contact and SJ area are reduced, and the SJ belt becomes patchy too. (**F**) Quantification of SJ surface area (green column) and the contact area (grey column), and the ratio between these two area (black point) under different PKA activity levels, ± SEM, n = 15–21. (**G**) Quantification of the mean length of individual SJ segments (green) and the mean total length of SJs (blue) under different PKA activity levels, measured in random nerve cord sections, ± SEM, n = 9–92. Asterisks in (**F, G**) indicate significance levels of comparisons based on Welch's ANOVA with Dunnett's T3 multiple comparisons test, n.s., p>0.05; *p<0.05; **p<0.01, ***p<0.001. Compared to WT, the mean total length of SJs significantly increases (4.28 ± 0.43 µm vs. 2.16 ± 0.14 µm, p=0.000523, about twofold) and the mean length of individual SJ segments significantly decreases (0.69 ± 0.08 µm vs. 2.16 ± 0.14 µm, p<0.0001, about 0.3-fold) under PKA overactivity; both the mean total length of SJs (1.49 ± 0.08 µm vs. 2.16 ± 0.14 µm, p=0.000878, about 0.69-fold) and the mean length of individual SJ segments decrease (0.67 ± 0.14 µm vs. 2.16 ± 0.14 µm, p<0.0001, about 0.31-fold), respectively, upon reduced PKA activity.

The online version of this article includes the following figure supplement(s) for figure 4:

**Figure supplement 1.** Morphology of apical membrane protrusions of subperineural glia (SPG) at different protein kinase A (PKA) activity levels, as revealed by serial section transmission electron microscopy (ssTEM).

Finally, SPG send apical protrusions into the neural cortex (*Figure 4—figure supplement 1*). These protrusions are much longer (2.01 ± 0.01 µm vs. 1.47 ± 0.09 µm, p=0.000230) upon elevated PKA activity and shorter than in WT (0.68 ± 0.09 µm vs. 1.47 ± 0.09 µm, p<0.0001) upon reduced PKA activity, suggesting that PKA activity more generally controls membrane protrusions and extension (*Figure 4—figure supplement 1*).

Taken together, our ultrastructural analyses and new 3D models support the light microscopic findings, and they provide superior quantification of the relevant parameters. Importantly, cell contact and SJ area, as well as total SJ content, are monotonically correlated with PKA activity, while individual SJ segment length is not. This suggests that the discontinuity of the SJ belt is the main cause for the observed BBB permeability defects.

## The Moody/PKA signaling pathway is polarized in SPG

The SPG are very thin cells, measuring around 0.2 µm along the apical–basal axis. In the embryo, the hemolymph-facing basal surface of the SPG is covered by a basal lamina (*Fessler et al., 1994*; *Olofsson and Page, 2005*; *Tepass and Hartenstein, 1994*), while during larval stages, the perineurial glia (PNG) form a second sheath directly on top of the SPG epithelium, which then serves as the basal contact for the SPG (*Stork et al., 2008*; *Stork et al., 2008*). Consistent with its chemoprotective function, the Mdr65 transporter localizes to the hemolymph-facing basal surface of the SPG, while Moody localizes to the CNS-facing apical surface (*Mayer et al., 2009*). The shallow lateral compartment contains the SJs, which not only seal the paracellular space but also act as a fence and prevent diffusion of transmembrane proteins across the lateral compartment. The apical localization of Moody protein is dependent on the presence of SJs (*Schwabe et al., 2017*).

To visualize the subcellular protein distributions along the apical–basal axis, we labeled them together with the SPG nuclei (*moody>nucCherry*). We examined the subcellular distribution of PkaC1 by immunohistochemistry (anti-PKA catalytic subunit antibody, which only bind to the catalytic subunits of PKA dissociated from the regulatory subunits of PKA after cAMP activation, not binding to the inactive enzyme) and found that active PkaC1 is enriched on the basal side of the SPG, and thus the opposite of the apically localized Moody (*Figure 5A*). This result is intriguing given PKA's antagonistic role in Moody signaling and suggests that pathway activity may affect the localization of pathway components.

We further examined the subcellular distribution of Moody and Pka-C1 in gain- or loss-of-function conditions of Moody/PKA signaling. Notably, the subcellular distribution of PkaC1 was indeed altered when Moody is knocked down in SPG (moody>Moody RNAi). PkaC1 lost its basal intracellular localization and appeared spread out throughout the cytoplasm (*Figure 5B*), suggesting that Moody is required for Pka-C1 polarized localization. Upon reduced PKA activity, Moody loses its apical localization (*Figure 5C*). Meanwhile, upon increased PKA activity, Pka-C1 appears at both sides of SPG as expected (*Figure 5E*), and Moody loses its apical membrane localization and is found at both the apical and basal membranes of SPG (*Figure 5E*), suggesting that PKA activity affects the subcellular localization of Moody. In addition, under gain of function of Moody signaling (moody>Gαo-GTP), PKA

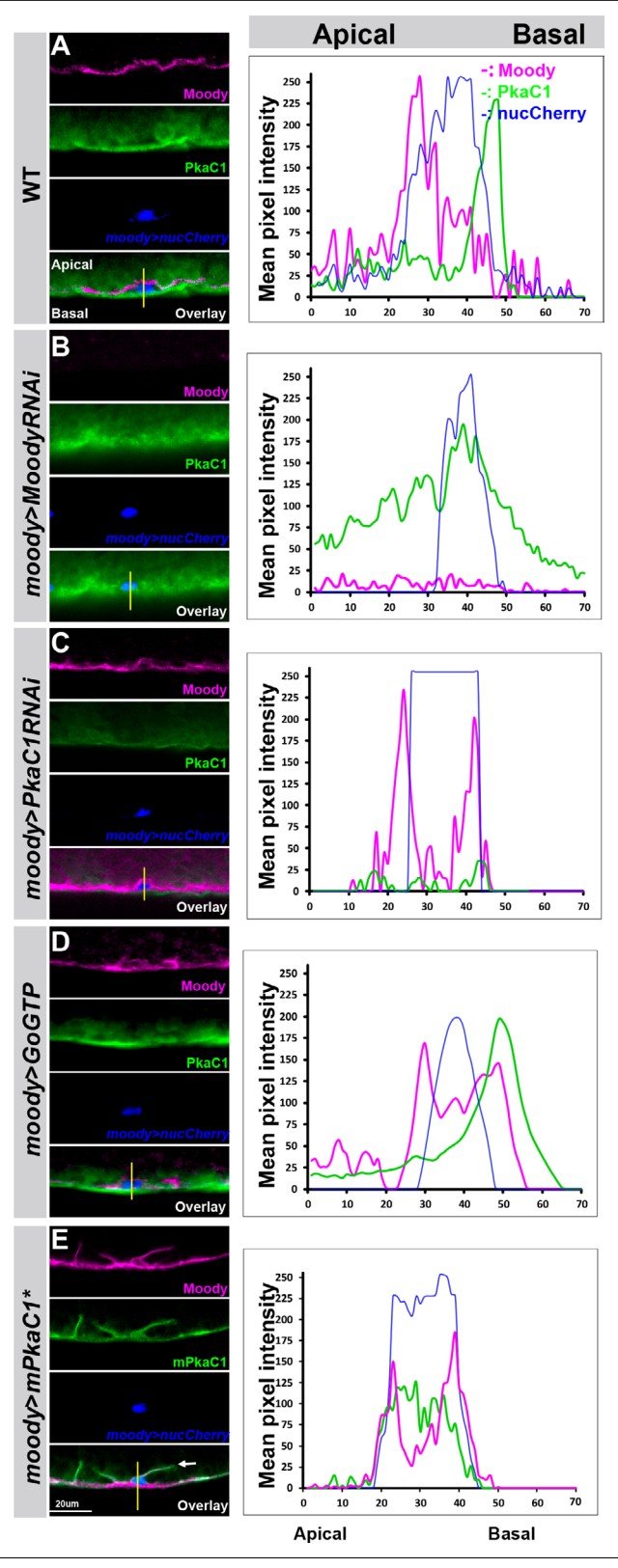

**Figure 5.** The Moody/protein kinase A (PKA) signaling pathway is polarized in subperineural glia (SPG). The subcellular localization of the PKA catalytic subunit PkaC1 and Moody in SPG of third instar larvae in WT (**A**), Moody knockdown (*moody>MoodyRNAi*) (**B**), PkaC1 knockdown (*moody>*Pka-C1-RNAi), GPCR gain of function (*moody>*Go GTP), and PKA overexpression (*moody>*mPka-C1*). Antibody labeling of Moody (magenta), of

*Figure 5 continued on next page*

*Figure 5 continued*

*Drosophila* PkaC1 or mouse PkaC1 (green), and of SPG nuclei (*moody>nucCherry*; blue). (**A–E**) Lateral views of the CNS/hemolymph border, with CNS facing top. On the right column, line scans of fluorescence intensities for each channel along the apical–basal axis at the positions indicated. In WT (**A**), Moody localizes to the apical side and PkaC1 is enriched at the basal side of SPG. Under loss of Moody signaling (*moody>MoodyRNAi*) (**B**), PKA spreads throughout the cell and loses its basal localization. Moody loses its apical localization under reduced (**C**) or increased PKA activity (**E**). Under GPCR gain of function (**D**), Pka-C1 is basally polarized, while Moody lost its asymmetric localization in SPG.

---

remains basally localized (*Figure 5D*), regardless of Moody's mis-localization, suggesting that Gαo signaling is sufficient for PKA enrichment at the basal side of SPG. Given the profound effect of PKA activity on the organization of SJ belt (*Figure 2B and D*), the cytoskeleton architecture and polarity, as well as vesicle transport (*Figure 3A–F*), mis-localized Moody could be an effect of deregulated protein trafficking or dysfunction of SJs, which normally restrict the diffusion of molecules between membrane compartments. Taken together, our data suggest that apical Moody signaling is necessary for repressing apical PkaC1 protein accumulation, and that this polarized subcellular localization results from the antagonistic relationship between Moody and PKA.

## MLCK and Rho1 function as PKA targets in the SPG

Considering that the most pronounced effect of increasing PKA levels in the SPG is a commensurate increase in membrane overlap at the basolateral side, we sought to genetically identify PKA targets involved in this process. PKA is known to regulate actomyosin contractility by phosphorylating and inhibiting myosin light chain kinase (MLCK), which leads to a decrease in Myosin light chain (MLC) phosphorylation and a concomitant reduction of actomyosin contractility in cell migration and endothelial barrier (*Garcia et al., 1995*; *Garcia et al., 1997*; *Howe, 2004*; *Tang et al., 2019*; *Verin et al., 1998*). To determine whether MLCK is required for BBB function, we examined two MLCK zygotic mutants, $MLCK^{02860}$ and $MLCK^{C234}$, and detected moderate BBB permeability in the late embryo (*Figure 6A* and *Figure 6—figure supplement 1*), indicating that MLCK plays a role in CNS insulation. Next, we asked whether PKA and MLCK function in the same signaling pathway using dominant genetic interaction experiments. We found that the BBB permeability of $PkaC1^{B3}$ heterozygous mutants could be rescued by removing one parental copy of *MLCK* ($MLCK^{02860}$ or $MLCK^{C234}$; *Figure 6B*). This suggests that MLCK interacts with PkaC1 in the SPG. Finally, we examined BBB insulation and SJ defects of *MLCK* zygotic mutant larva ($MLCK^{C234}$). $MLCK^{C234}$ mutant larvae showed significant BBB permeability and a widened SJ belt (*Figure 6C–F*) compared to WT (*Figure 2B*), but the phenotypes were milder than those of PKA overactivity (*Figure 2B*).

PKA is also known to phosphorylate and inhibit the small GTPase Rho1, which reduces the activity of its effector Rho kinase (ROK), ultimately resulting in decreased MLC phosphorylation and actomyosin contractility (*Dong et al., 1998*; *Garcia et al., 1999*; *Lang et al., 1996*; *Tang et al., 2019*; *Xu and Myat, 2012*). Moreover, RhoA activity has been shown to drive actin polymerization at the protrusion of migrating cells (*Machacek et al., 2009*), and a PKA-RhoA signaling has been suggested to act as a protrusion-retraction pacemaker at the leading edge of the migrating cells (*Tkachenko et al., 2011*). To check if Rho1 is required for BBB function, we determined the BBB permeability in the late embryo and third instar larval stages. Two loss-of-function alleles, the hypomorphic allele $Rho1^{1B}$ (*Magie and Parkhurst, 2005*) and the null allele $Rho1^{E.3.10}$, showed dye penetration defects as homozygous zygotic mutant embryos, with the null allele showing a particularly pronounced effect (*Figure 6B* and *Figure 6—figure supplement 1*). At the larval stage, the SPG-specific Rho1 knockdown (*moody>Rho1* RNAi) resulted in strong dye penetration into the nerve cord (*Figure 6C and E*). These results suggest that Rho1 is required for the formation and continued growth of the BBB. We again asked whether PKA and Rho1 function in the same pathway and performed dominant genetic interaction experiments using a sensitized genetic background. The embryonic dye penetration defects of PkaC1 heterozygous mutants ($PkaC1^{B3}$) were significantly reduced by removing one genomic copy of the Rho1 null allele ($Rho1^{E.3.10}$), but not by removing one copy of the hypomorphic allele $Rho1^{1B}$ (*Figure 6B*). These findings suggest that Rho1 is a PKA target in BBB regulation. Collectively, our results indicate that PKA suppresses actomyosin contractility in a two-pronged fashion by negatively regulating both MLCK and Rho1.

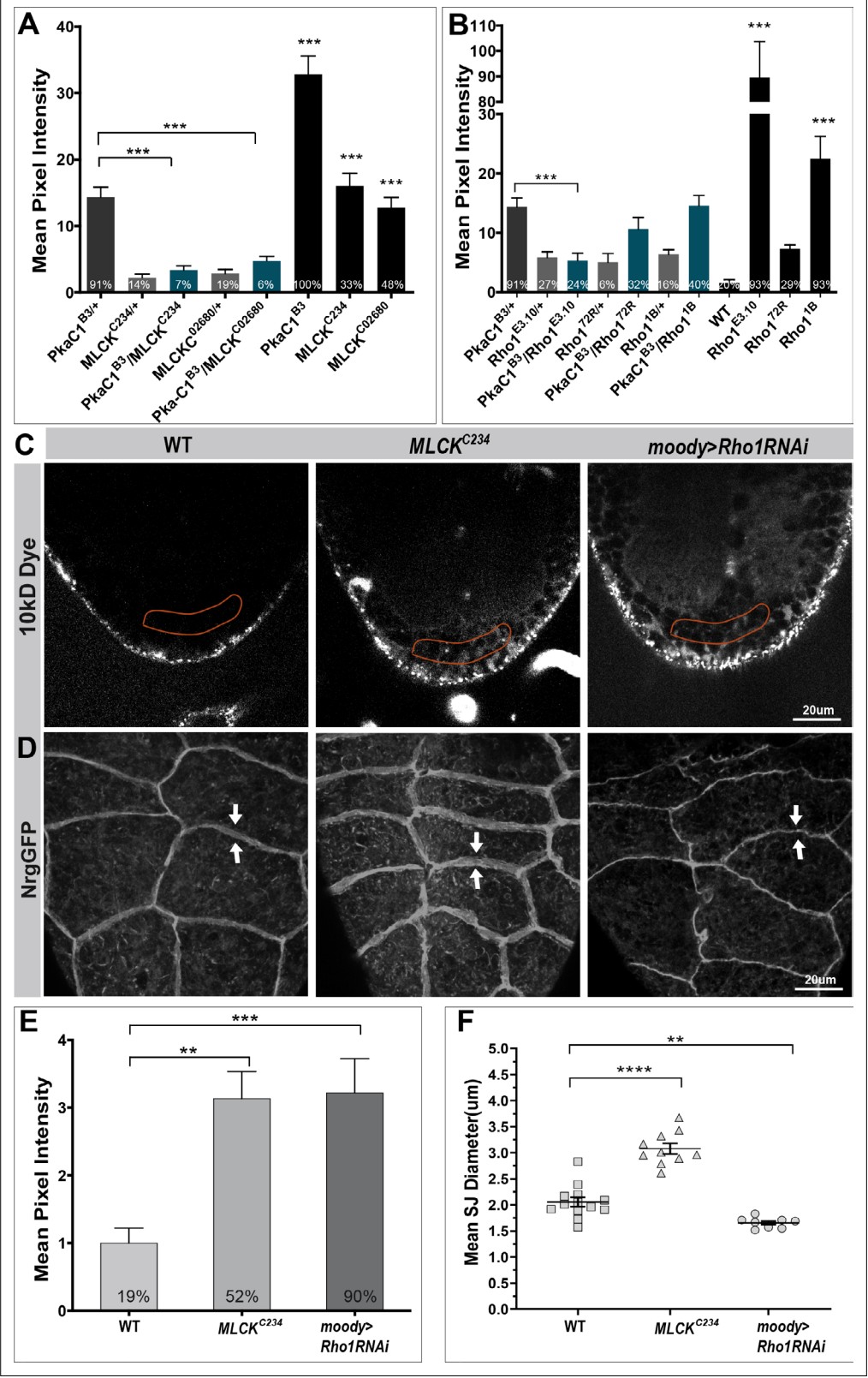

**Figure 6.** Myosin light chain kinase (MLCK) and Rho1 function as protein kinase A (PKA) targets in subperineural glia (SPG). (**A**) Quantification of dye penetration effects in the embryo of *MLCK* and *Rho1*. (**B**) Dominant genetic interactions between *PkaC1^B3* and *MLCK* and *Rho1* mutant heterozygotes as quantified by dye penetration in the embryo. In (**A, B**), columns represent the strength of dye penetration into the nerve cord as measured by the

*Figure 6 continued on next page*

*Figure 6 continued*

mean pixel intensity, ± SEM, n = 14–98. (**C, D**) Blood–brain barrier (BBB) phenotype of *MLCK* zygotic mutant and SPG-specific Rho1 knockdown (*moody>Rho1* RNAi) animals in single confocal sections of dye injected third instar larvae (**C**), and septate junction (SJ) morphology using the NrgGFP marker (**D**), with width of SJ belt highlighted by arrows. (**E**) Quantification of the dye penetration assay from (**C**). Columns represent intensity of dye penetration as measured by mean pixel intensity and normalized to WT mean (see *Materials and methods*, ± SEM, n = 13–19). (**F**) Quantification of the mean diameter of SJ belts from (**D**), ± SEM, n = 8–13. In (**A**), (**B**), and (**E**), the percentage of animals showing dye penetration is indicated at the bottom of each column. Asterisks in (**A**), (**B**), (**E**), and (**F**) indicate significance levels of comparisons against either WT in (**E**) and (**F**) or *PkaC1^{B3}* group in (**A**) and (**B**) based on Brown–Forsythe and Welch's ANOVA with multiple comparisons test, n.s., $p > 0.05$; *$p < 0.05$; **$p < 0.01$, ***$p < 0.001$.

The online version of this article includes the following figure supplement(s) for figure 6:

**Figure supplement 1.** Myosin light chain kinase (MLCK) and Rho1 are required for blood–brain barrier (BBB) integrity in embryos.

## Discussion

Previous studies implicated a novel GPCR signaling pathway in the formation of the *Drosophila* BBB in late embryos (*Bainton et al., 2005*; *Schwabe et al., 2005*). This work also revealed that besides the GPCR Moody two heterotrimeric G proteins (Gαiβγ, Gαoβγ) and the RGS Loco participate in this pathway. Here we provide a comprehensive molecular and cellular analysis of the events downstream of G protein signaling using a candidate gene screening approach. We present new, more sensitive methods for phenotypic characterization and extended the analysis beyond the embryo into larval stages. This work identifies PKA, together with some of its targets, as crucial antagonistic effectors in the continued cell growth of SPG and maintenance of the BBB sealing capacity. This role is critical to ensure proper neuronal function during BBB formation and maturation.

Multiple lines of evidence demonstrate a role of PKA for proper sealing of the BBB: loss of PKA activity leads to BBB permeability defects, irregular growth of SPG during epithelium formation, reduced membrane overlap, and a narrower SJ belt at SPG cell–cell contacts. The role of PKA as an effector of the Moody signaling pathway is further supported by dominant genetic interaction experiments, which show that the dye penetration phenotype of *PkaC1* heterozygous mutant embryos was partially rescued by removing one genomic copy of *Gβ13F* or *loco*. Moreover, the analysis of the larval phenotype with live SJ and cytoskeleton markers shows that PKA gain of function behaved similarly to Moody loss of function. Conversely, PKA loss of function resembled the overexpression of GαoGTP, which mimics Moody gain-of-function signaling.

Our results from modulating PKA activity suggest that the total cell contact and SJ areas are a major function of PKA activity: low levels of activity cause narrow contacts, and high levels give rise to broad contacts. Moreover, the analysis of various cellular markers (actin, microtubules, SJs, vesicles) indicates that the circumferential cytoskeleton and delivery of SJ components respond proportionately to PKA activity. This, in turn, promotes the changes in cell contact and junction areas coordinately at the lateral side of SPG. Our experiments demonstrate that the modulation of the SPG membrane overlap by PKA proceeds, at least in part, through the regulation of actomyosin contractility, and that this involves the phosphorylation targets MLCK and Rho1. This suggests that crucial characteristics of PKA signaling are conserved across eukaryotic organisms (*Bauman et al., 2004*; *Marks and Kalderon, 2011*; *Park et al., 2000*; *Taylor et al., 1990*).

At the ultrastructural level, our ssTEM analysis of the larval SPG epithelium clarifies the relationship between the inter-cell membrane overlaps and SJ organization and function. Across different PKA activity levels, the ratio of SJ areas to the total cell contact area remained constant at about 30% . This proportionality suggests a mechanism that couples cell contact with SJ formation. The primary role of Moody/PKA appears in this process to be the control of membrane contacting area between neighboring cells. This is consistent with the results of a temporal analysis of epithelium formation and SJ insertion in late embryos of WT and Moody pathway mutants, which shows that membrane contact precedes and is necessary for the appearance of SJs (*Schwabe et al., 2017*). The finding that the surface area that SJs occupy did not exceed a specific ratio, irrespective of the absolute area of cell contact, suggests an intrinsic, possibly steric limitation in how much junction can be fitted into a given cell contact space. While most phenotypic effects are indeed a major function of Moody and PKA activity, the discontinuity and shortening of individual SJ strands is not. It occurred with both

increased and decreased signaling and appears to cause the leakiness of the BBB in both conditions. Our ssTEM-based 3D reconstruction thus demonstrates that the total area covered by SJs and the length of individual contiguous SJ segments are independent parameters. The latter appears to be critical for the paracellular seal, consistent with the idea that Moody plays a role in the formation of continuous SJ stands.

The asymmetric localization of PKA that we observed sheds further light on the establishment and function of apical–basal polarity in the SPG epithelium. Prior to epithelium formation, contact with the basal lamina leads to the first sign of polarity (*Schwabe et al., 2017*). Moody becomes localized to the apical surface only after epithelial closure and SJ formation, suggesting that SJs are required as a diffusion barrier and that apical accumulation of Moody protein is the result of polarized exocytosis or endocytosis (*Schwabe et al., 2017*). Here, we now show that the intracellular protein PKA catalytic subunit-PkaC1 accumulates on the basal side of SPG, and that this polarized accumulation requires (apical) Moody activity. Such an asymmetric, activity-dependent localization has not previously been described for PKA in endothelium, and while the underlying molecular mechanism is unknown, the finding underscores that generating polarized activity along the apical–basal axis of the SPG is a key element of Moody pathway function.

An intriguing unresolved question is how increased SPG cell size and SJ length can keep up with the expanding brain without disrupting the BBB integrity during larva growth. We found that the SJ grows dramatically in length (0.57 ± 0.07 μm vs. 2.16 ± 0.14 μm, about 3.7-fold) from the late embryo (*Figure 1E*) to third instar larva (*Figure 4G*), which matches the increased cell size of SPG (about four-fold; *Babatz et al., 2018*; *Unhavaithaya and Orr-Weaver, 2012*). During the establishment of the SPG epithelium in the embryo, both increased and decreased Moody signaling resulted in asynchronous growth and cell contact formation along the circumference of SPG, which in turn led to irregular thickness of the SJ belt (*Schwabe et al., 2017*). Therefore, a similar relationship may exist during the continued growth of the SPG epithelium in larvae, with the loss of continuity of SJ segments in Moody/PKA mutants resulting from unsynchronized expansion of the cell contact area and an ensuing erratic insertion of SJ components. Since SJs form relatively static complexes, any irregularities in their delivery and insertion may linger for extended periods of time (*Babatz et al., 2018*; *Deligiannaki et al., 2015*; *Oshima and Fehon, 2011*). The idea that shortened SJ segments are a secondary consequence of unsynchronized cell growth is strongly supported by our finding that disruption of actomyosin contractility in MLCK and Rho1 mutants compromises BBB permeability.

Collectively, our data suggest the following model: polarized Moody/PKA signaling controls the cell growth and maintains BBB integrity during the continuous morphogenesis of the SPG secondary epithelium. On the apical side, Moody activity represses PKA activity (restricting local cAMP level within the apial-basal axis in SPG) and thereby promotes actomyosin contractility. On the basal side, which first adheres to the basal lamina and later to the PNG sheath, PKA activity suppresses actomyosin contractility via MLCK and Rho1 phosphorylation and repression (*Figure 7*). Throughout development, the SPG grow continuously while extending both their cell surface and expanding their cell contacts. Our data suggest that the membrane extension occurs on the basolateral surface through insertion of plasma membrane and cell-adhesive proteins, with similar behavior in epithelial cell, but regulated by a distinct polarized Moody/PKA signaling in SPG (*Wojtal et al., 2008*). In analogy to motile cells, the basal side of the SPG would thus act as the 'leading edge' of the cell, while the apical side functions as the 'contractile rear' (*Nelson, 2009*). According to this model, Moody/Rho1 regulate actomyosin to generate the contractile forces at the apical side to driving membrane contraction, which directs the basolateral insertion of new membrane material and SJs. In this way, differential contractility and membrane insertion act as a conveyor belt to move new formed membrane contacts and SJ from the basolateral to apical side. Loss of Moody signaling leads to symmetrical localization of PKA and to larger cell contact areas between SPG due to diminished apical constriction. Conversely, loss of PKA causes smaller cell contact areas due to increased basal constriction.

Our results may have important implications for the neuron–glia interaction in the nervous system and the development and maintenance of the BBB in vertebrates. SJs have several structural and functional components in common with paranodal junctions, which join myelinating glial cells to axons in the vertebrate nervous system, and they share similar regulation mechanisms (*Hortsch and Margolis, 2003*; *Salzer, 2003*; *Salzer, 2015*). The vertebrate BBB consists of a secondary epithelium with interdigitations similar to the ones between the *Drosophila* SPG (*Chow and Gu, 2015*; *Cong and Kong,*

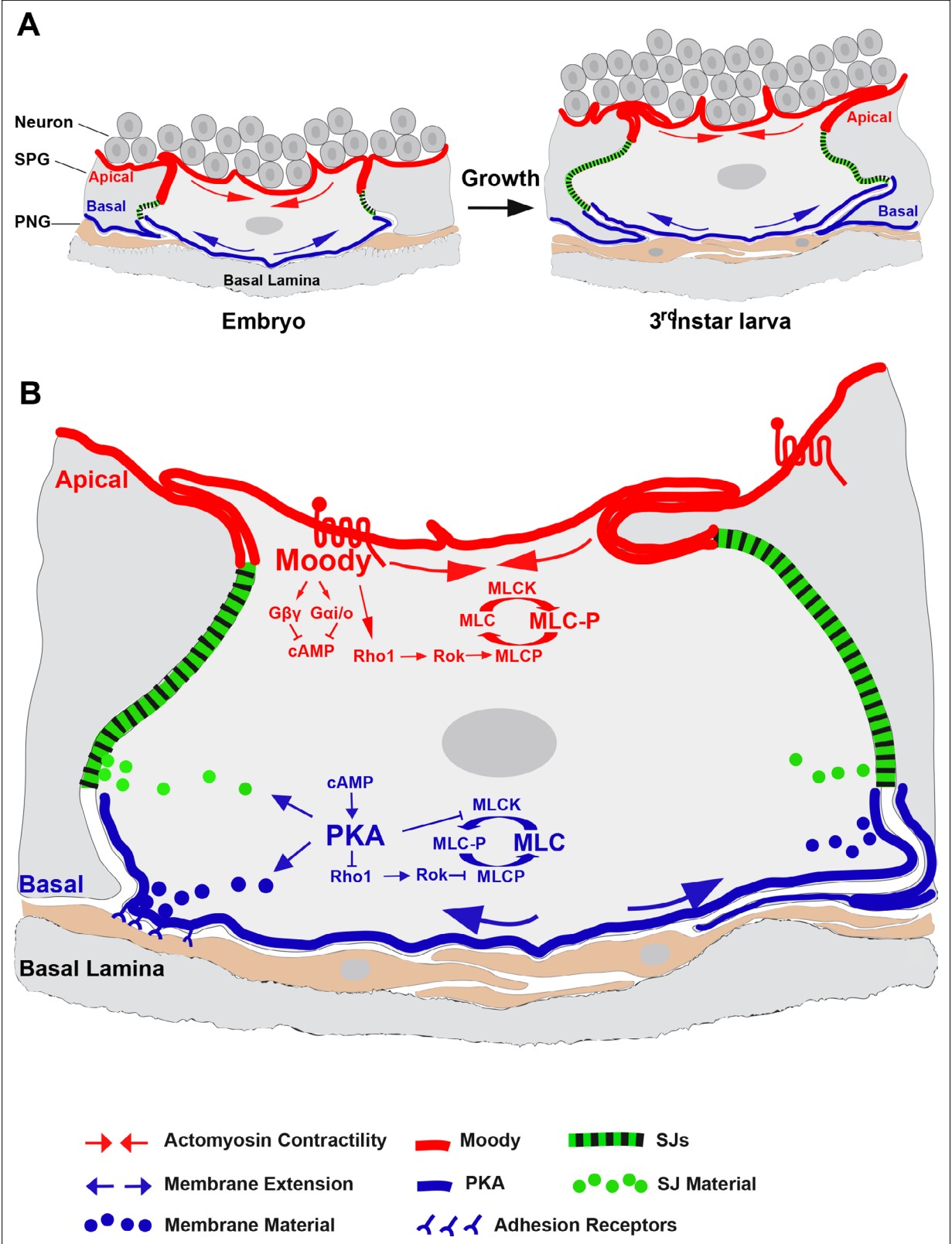

**Figure 7.** Model of Moody/protein kinase A (PKA) signaling in the glial blood–brain barrier (BBB). Schematic depicting polarized Moody/PKA signaling along the apical–basal axis and its cellular function in controlling subperineural glia (SPG) continued cell growth (**A**) and BBB integrity (**B**) by differentially regulating actomyosin contractility and septate junction (SJ) organization spatiotemporally. For detailed description, see Discussion.

2020; *Hindle and Bainton, 2014*; *Reinhold and Rittner, 2017*). While the sealing is performed by TJs, it will be interesting to investigate whether there are similarities in the underlying molecular and cellular mechanisms that mediate BBB function (*Sugimoto et al., 2020*).

# Materials and methods

**Key resources table**

| Reagent type (species) or resource | Designation | Source or reference | Identifiers | Additional information |
|---|---|---|---|---|
| Genetic reagent (*Drosophila melanogaster*) | $PkaC1^{H2}$ | Bloomington Drosophila Stock Center | BDSC:4101, RRID:BDSC_4101 | FlyBase symbol: Dmel\Pka-C1$^{H2}$ |
| Genetic reagent (*D. melanogaster*) | $PkaC1^{B3}$ | Bloomington Drosophila Stock Center | FBal0033955 | Dmel\Pka-C1$^{B3}$ |
| Genetic reagent (*D. melanogaster*) | $PkaC1^{A13}$ | Bloomington Drosophila Stock Center | FBal0033953 | Dmel\Pka-C1$^{A13}$ |
| Genetic reagent (*D. melanogaster*) | UASmPkaC1*(mC*) | D.Kalderon | | DC0 |
| Genetic reagent (*D. melanogaster*) | moodyGAL4 | T.Schwabe | FBtp0022847 | P{moody-GAL4.S} |
| Genetic reagent (*D. melanogaster*) | repoGAL4 | Bloomington Drosophila Stock Center V.Auld | BDSC:7415 FBst0007415 | P{GAL4}repo |
| Genetic reagent (*D. melanogaster*) | $Nrg^{G305}$ | Bloomington Drosophila Stock Center W.Chia | BDSC:6,844 FBal0147727 | Dmel\Nrg$^{G00305}$ |
| Genetic reagent (*D. melanogaster*) | UASGFPMoesin | D. Kiehart | FBtp0017306 | *UASmRFPMoesin* |
| Genetic reagent (*D. melanogaster*) | UASmRFPMoesin | T. Schwabe | FBtp0022846 | P{UAS-Moe.RFP} |
| Genetic reagent (*D. melanogaster*) | $G\beta13F^{\Delta1-96A}$ | F. Matsuzaki | FBal0128192 | Dmel\Gβ13F$^{\Delta1-96A}$ |
| Genetic reagent (*D. melanogaster*) | UAStauGFP | M. Krasnow | FBtp0012358 | P{UAS-tauGFP} |
| Genetic reagent (*D. melanogaster*) | UASG$_{\alpha o}$GTP | A. Tomlinson | FBal0183487 | Dmel\Gαo$^{GTP.UAS}$ |
| Genetic reagent (*D. melanogaster*) | $loco^{\Delta13}$ | C. Klämbt | FBal0096758 | Dmel\loco$^{\Delta13}$ |
| Genetic reagent (*D. melanogaster*) | $moody^{\Delta17}$ | R. Bainton | FBab0044985 | Df(1)moody-Δ17 |
| Genetic reagent (*D. melanogaster*) | moody-RNAi | R. Bainton | FBtp0022779 | P{UAS-moody.dsRNA} |
| Genetic reagent (*D. melanogaster*) | UASnucmCherry | T. Schwabe | | P{UAS-nuc.Cherry} |
| Genetic reagent (*D. melanogaster*) | UASGFPEB1 | Bloomington Drosophila Stock Center D.Brunner | BDSC:35512 | Dmel\P{UAS-EB1-GFP}3 |
| Genetic reagent (*D. melanogaster*) | UASGFPNod | Bloomington Drosophila Stock Center | FBtp0014112 | P{UASp-nod-GFP} |
| Genetic reagent (*D. melanogaster*) | UASGFPRho | Bloomington Drosophila Stock Center | BDSC:9528 FBal0189974 | Dmel\Rho1$^{GFP}$ |
| Genetic reagent (*D. melanogaster*) | UASRab4RFP | Bloomington Drosophila Stock Center | BDSC:8505 FBtp0018526 | 9,562 |

*Continued on next page*

*Continued*

| Reagent type (species) or resource | Designation | Source or reference | Identifiers | Additional information |
|---|---|---|---|---|
| Genetic reagent (*D. melanogaster*) | *UASactinGFP* | Bloomington Drosophila Stock Center | BDSC:9562 FBtp0001557 | P{ActGFP} |
| Genetic reagent (*D. melanogaster*) | *Rho72R* | Bloomington Drosophila Stock Center | FBal0061660 | Dmel\Rho1^72R^ |
| Genetic reagent (*D. melanogaster*) | *Rho1B* | Bloomington Drosophila Stock Center | BDSC:9477 FBal0176027 | Dmel\Rho1^1B^ |
| Genetic reagent (*D. melanogaster*) | *MLCK02860* | Bloomington Drosophila Stock Center | BDSC:11089 FBal0159721 | Dmel\Strn-Mlck^c02860^ |
| Genetic reagent (*D. melanogaster*) | *MLCKC234* | Bloomington Drosophila Stock Center | BDSC:16314 FBal0159722 | Dmel\Strn-Mlck^C234^ |
| Genetic reagent (*D. melanogaster*) | *tubGAL80ts* | Bloomington Drosophila Stock Center | BDSC:7019 FBst0007019 | Dmel\P{tubP-GAL80^ts^}20 |
| Genetic reagent (*D. melanogaster*) | *PkaC1KK10896* | VDRC | VDRC:109758 | FBgn0037103 |
| Genetic reagent (*D. melanogaster*) | *Rho1KK108182* | VDRC | VDRC:v109420 FBal0259627 | Dmel\Rho1^KK108182^ |
| Genetic reagent (*D. melanogaster*) | *TauGD8682* | VDRC | VDRC:v25023 FBal0210735 | Dmel\tau^GD8682^ |
| Antibody | Anti-PkaC1 (rabbit polyclonal) | Lane ME; Genes Dev. 1993 | ABS571, RRID:AB_2568479 RRID:AB_2568479 | IF(1:400) |
| Antibody | Anti-PkaC1 (mouse monoclonal) | Abcam | ab15051, RRID:AB_2269474 | IF(1:500) |
| Antibody | Anti-REPO (mouse monoclonal) | Developmental Studies Hybridoma Bank | Cat#: 17-9987-42; RRID:AB_2043823 | IF(1:10) |
| Antibody | Anti-GFP (mouse monoclonal) | Molecular Probes | A-11120 | IF(1:100) |
| Antibody | Anti-Mega (mouse monoclonal) | R. Schuh | | IF(1:100) |
| Antibody | Anti-dContactin (guinea pig polyclonal) | M. Bhat | | IF(1:100) |
| Antibody | Anti-RFP (rabbit polyclonal) | US Biological | | IF(1:100) |
| Antibody | Anti-Moody b (rabbit polyclonal) | *Bainton et al., 2005*; *Schwabe et al., 2005* | | IF(1:500) |
| Software, algorithm | Fiji | NIH | | ImageJ |
| Software, algorithm | Imaris 4.0 | Bitplane | | |
| Chemical compound, drug | Texas red-coupled dextran, 10 kDa | Molecular Probes | D1828 | 10 mg/ml |

## Fly strains and constructs

The following fly strains were obtained from published sources: *PkaC1^H2^* (BDSC Cat# 4101, RRID:BDSC_4101); *PkaC1^B3^*; *PkaC1^A13^*; *UASmPkaC1\*(mC\*)* (D. Kalderon); *moodyGAL4* (T. Schwabe); *repoGAL4* (V. Auld); *Nrg^G305^* (*NrgGFP*; W.Chia); *UASGFPMoesin* (D.Kiehart); *UASmRFPMoesin* (T. Schwabe); *Gβ13F^Δ1-96A^* (F. Matsuzaki); *UAStauGFP* (M. Krasnow); *UASG_αo_GTP* (A. Tomlinson), *loco^Δ13^* (C. Klämbt); *moody^Δ17^*(R. Bainton); *moody-RNAi* (R. Bainton); *UASnucmCherry* (T. Schwabe); *UASGFPEB1* (D. Brunner); *UASGFPNod, UASGFPRho, UASactinGFP, UASRab4RFP, Rho^72R^, Rho^1B^, MLCK^02860^, MLCK^C234^, tubGAL80^ts^* (Bloomington Stock Center); *PkaC1^KK108966^, Rho1^KK108182^, Tau^GD8682^*(-VDRC). For live genotyping, mutant and transgenic lines were balanced (*Kr::GFP*) (*Casso et al., 1999*) or positively marked using *nrgNrgGFP*. Temperature-sensitive control of gene expression in

eLife Research article

Cell Biology | Developmental Biology

SPG is achieved by using a *tubGAL80ts; moodyGAL4* driver. All strains were raised at 25 °C. except for *tubGAL80ts; moodyGAL4* crosses, which were raised at 18 °C until 1 day after eclosion and then shifted to 29°.

## Live imaging

Dissected third instar larval cephalic complexes were mounted in PBS and imaged directly. All confocal images were acquired using a Zeiss LSM 510 or 710 system. Stacks of 20–40 0.5- µm confocal sections were generated; image analysis was performed using Zeiss LSM 510, ImageJ (NIH) or Imaris 4.0 (Bitplane) software. The results for each section were assembled as a separate channel of the stack. Time-lapse recordings were carried out on 12 hr *after egg lay (AEL)* embryos raised at 20 °C using an inverted Zeiss LSM 510 confocal microscope. To increase signal strength, the pinhole was opened to 1.3 (z-section thickness 0.6 µm), and z-stacks of 12 sections were acquired once per minute. To adjust for focus drift, which is mainly caused by rotation of the embryo, the z-stack coordinates were adjusted at various timepoints without disrupting the continuity of the movie. Between 5 and 7 movies were captured per genotype, each 80–110 min in duration.

## Immunohistochemistry

Immunohistochemistry was performed following standard procedures (*Bainton et al., 2005*; *Schwabe et al., 2005*). The antibodies used in the study were rabbit α-PkaC1 (1:400, Pka-C1, RRID:AB_2568479; *Lane and Kalderon, 1993*), mouse α-PkaC1 (1:100, BD), mouse α-REPO (1:10, Developmental Studies Hybridoma Bank), mouse α-GFP (1:100, Molecular Probes), mouse α-Mega (1:100, R. Schuh), guinea pig α-dContactin (1:1000, M. Bhat), and rabbit α-RFP (1:100, US Biological). Fluorescent secondary antibodies were coupled to Cy3 (1:500, Jackson), Alexa Fluor 488 or Alexa Fluor 633 (1:500, Molecular Probes). Rat α-Moody β was generated in the lab (1:500; *Bainton et al., 2005*; *Schwabe et al., 2005*).

## Image analysis

The width of the SJ belt was extracted from maximum intensity projections (MIPs) along the z-axis of 3D confocal stacks of the nervous system. Specifically, we used Imaris 4.0 to perform 2D segmentation of the GFP-marked SJs. For each of the markers, an optimal threshold for the pixel intensity was chosen by fitting the obtained segmented pattern with the raw fluorescence signal. To evaluate the average thickness of the SJs, we splitted the SJ segments into sections of 3–4 µm in length. An approximation of the diameters of the single sections was then obtained by extracting their ellipticity parameters along the axis perpendicular to their main axis. A mean diameter of the SJ was calculated by averaging over the diameters of all single sections. For quantification, random images were chosen with each marker and the distribution of all live markers was measured by Fiji software. To calculate the changes of the distribution of markers, plot profiles of fluorescence intensities along the nucleus to the membrane in each cell were divided into two compartment, membrane area and nucleus area (half to half). Then we calculated the ratio of mean pixel intensity between these two areas (membrane/nucleus) under different PKA activity with each marker. The statistical analysis was performed using Brown–Forsythe and Welch's ANOVA with multiple comparisons test.

## Dye-penetration assay in embryo, third instar larva, and adult flies

The dye penetration assay in embryos was performed as described (*Schwabe et al., 2005*). For the dye penetration assay in third instar larvae, a fluorescent dye (Texas red-coupled dextran, 10 kDa, 10 mg/ml, Molecular Probes) was injected into the body cavity of third instar larva. After 2.5 hr, the cephalic complex was dissected, and the dye penetrated into the nerve cord was analyzed using Zeiss LSM710 confocal microscopy. Dye penetration was quantified by calculating the percentage of larva showing dye penetration and by measuring the mean pixel intensity within a representative window of the ventral portion of the nerve cord using Fiji software, and normalized by dividing by the mean of the WT control group. To assess the significance of effects for the embryonic and larval dye penetration assays, Brown–Forsythe and Welch's ANOVA with multiple comparisons test was performed.

The dye penetration assay in adult flies was performed as described in *Bainton et al., 2005* with some critical modifications. Briefly, adult flies were hemolymph injected with 10 mg/ml 10 kDa Texas red-coupled dextran. After 2 hr, the injected flies were decapitated and their heads were mounted in a fluorinated grease-covered glass slides with two compound eyes on the side (the proboscis facing up).

Images were acquired on a Zeiss LSM710 confocal microscope at 200–300 µm depths from the eye surface with a Plan Fluor 10xw objective. Dye penetration was quantified by measuring the mean pixel intensities within a representative window of the central region of retina (n = 18–30) of maximum-intensity Z projection of each image stack (z-section thickness 0.6 µm) by Fiji software and normalized by the WT control. Statistical significance was assessed using the two-tailed unpaired t-test.

### Transmission electron microscopy (TEM)

Late stage 17 (22–23 hr AEL) embryos were processed by high-pressure freezing in 20% BSA, freeze-substituted with 2% $OsO_4$, 1% glutaraldehyde, and 0.2% uranyl acetate in acetone (90%), $dH_2O$ (5%), methanol (5%) over 3 days (–90°C to 0°C), washed with acetone on ice, replaced with ethanol, infiltrated and embedded in Spurr's resin, sectioned at 80 nm and stained with 2% uranyl acetate and 1% lead citrate for 5 min each. Sections were examined with a FEI TECNAI G2 Spirit BioTwin TEM with a Gatan 4K x 4K digital camera. For quantification, random images were shot, and the length of visible SJ membrane stretches in each image was measured using Fiji software. Statistics were calculated using the two-tailed unpaired t-test.

### Serial section transmission electron microscopy (ssTEM)

Freshly dissected third instar larval CNSs were fixed in 2% glutaraldehyde and 2% $OsO_4$ in 0.12 M sodium cacodylate (pH 7.4) by microwave (Ted Pella, BioWave Pro MW) as follows: 30" at 300 W, 60" off, 30" at 350 W; 60" off, 30" at 400 W. The samples were then rinsed 2 × 5′ with cold 0.12 M sodium cacodylate buffer; post-fixed with 1% $OsO_4$ in 0.12 M sodium cacodylate buffer (pH 7.4) on an ice bath by microwave as follows: 30" at 350 W, 60" off, 30" at 375 W, 60" off, 30" at 400 W; rinsed 2 × 5′ with 0.12 M sodium cacodylate buffer at RT; 2 × 5′ with distilled water at RT; stained in 1% uranyl acetate overnight in 4 °C; rinsed 6 × 5′ with distilled water; dehydrated with ethanol followed by propylene oxide (15′); infiltrated and embedded in Eponate 12 with 48 hr polymerization in a 65 °C oven. 50 nm serial sections were cut on a Leica UC6 ultramicrotome and picked up with Synaptek slot grids on a carbon-coated Pioloform film. Sections were post-stained with 1% uranyl acetate followed by Sato's (1968) lead. The image acquisition of multiple sections (~150 sections in each genotype) and large tissue areas was automatically captured with a Gatan 895 4K × 4K camera by a FEI Spirit TECNAI BioTWIN TEM using Leginon (*Suloway et al., 2005*). TrakEM2 software was used to montage, align images, trace, and reconstruct 3D SJ structures between contacting SPG within and across serial sections. For quantification, random images were chosen, and the length of visible SJs stretches and membrane contacting area in each image was measured using Fiji. The statistical analysis was performed using Brown–Forsythe and Welch's ANOVA with multiple comparisons test.

## Acknowledgements

We thank D Kalderon, R Bainton, G Beitel, V Auld, M Bhat, W Chia, Y Hiromi, M Hortsch, B Jones, D Kiehart, C Klämbt, J Knoblich, M Krasnow, M Peifer, A Tomlinson, R Tsien, and the Developmental Studies Hybridoma Bank for providing us with fly strains, constructs, and antibodies. Special thanks go to U Unnerstall, M Deligiannaki, A Caspe, Schroeder, S Axelrod, E Kurant, and Li WH for their helpful comments on the manuscript. We are grateful to all members of the Gaul and Steller labs for their continued support of this work. This work was supported by an Alexander von Humboldt-Professorship from the Bundesministerium für Bildung und Forschung (UG), the Center for Integrated Protein Science (UG), the NIH (5R01EY011560) (UG, XL), UG acknowledges support by the Deutsche Forschungs gemeinschaft (SFB 646, SFB 1064, CIPSM, QBM) and the Bundesministerium für Bildung und Forschung (Alexander von Humboldt-Professorship, BMBF: ebio).

## Additional information

### Funding

| Funder | Grant reference number | Author |
|---|---|---|
| National Institutes of Health | 5R01EY011560 | Xiaoling Li Ulrike Gaul |
| Deutsche Forschungsgemeinschaft | SFB 646 | Ulrike Gaul |
| Bundesministerium für Bildung und Forschung | Alexander von Humboldt-Professorship | Ulrike Gaul |
| Deutsche Forschungsgemeinschaft | SFB1064 | Ulrike Gaul |
| Deutsche Forschungsgemeinschaft | SFB 1064 | Ulrike Gaul |

The funders had no role in study design, data collection and interpretation, or the decision to submit the work for publication.

### Author contributions

Xiaoling Li, Conceptualization, Data curation, Formal analysis, Funding acquisition, Investigation, Methodology, Project administration, Software, Supervision, Validation, Visualization, Writing - original draft, Writing – review and editing; Richard Fetter, Investigation, Writing – review and editing; Tina Schwabe, Methodology, Writing – review and editing; Christophe Jung, Formal analysis, Methodology; Liren Liu, Formal analysis, Resources, Writing – review and editing; Hermann Steller, Conceptualization, Project administration, Resources, Supervision, Writing – review and editing; Ulrike Gaul, Conceptualization, Data curation, Funding acquisition, Project administration, Resources, Supervision, Writing – review and editing

### Author ORCIDs

Xiaoling Li http://orcid.org/0000-0002-3408-7066
Hermann Steller http://orcid.org/0000-0002-4577-4507

### Decision letter and Author response

Decision letter https://doi.org/10.7554/eLife.68275.sa1
Author response https://doi.org/10.7554/eLife.68275.sa2

## Additional files

### Supplementary files
• Transparent reporting form

### Data availability

All data generated or analysed during this study are included in Dyrad generic databases with DOI https://doi.org/10.5061/dryad.fj6q573tx. Source data files have been provided for Figures 1, 2, 3, 4, 6 and Figure supplement 2 and 5.

The following dataset was generated:

| Author(s) | Year | Dataset title | Dataset URL | Database and Identifier |
|---|---|---|---|---|
| Li X, Fetter R, Schwabe T, Jung C, Liu L, Steller H, Gaul U | 2021 | The cAMP effector PKA mediates Moody GPCR signaling in *Drosophila* blood-brain barrier formation and maturation | https://doi.org/10.5061/dryad.fj6q573tx | Dryad Digital Repository, 10.5061/dryad.fj6q573tx |

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
