## [Decision Letter]

**Acceptance summary:**

This elegant work investigates the structure and maintenance of the blood brain barrier (BBB) in *Drosophila*. Understanding how the BBB is established and functions is relevant to all animals with a BBB including humans, where it is critically important for regulating drug/small molecule access to the brain. Previous work from this lab and others have shown that the BBB is composed of a specialized type of glia called the subperineurial glia (SPG), which enwrap the entire central nervous system (CNS). Furthermore, they previously identified the Moody G protein coupled receptor (GPCR) as being specifically expressed in SPG and required for BBB formation and maintenance. Here they show that Moody protein is localized to the apical membrane domain (facing the CNS) while Protein Kinase A (PKA) is localized to the complementary basal membrane domain (facing the hemolymph of the body cavity). Not only do Moody and PKA have non-overlapping subcellular localization, but genetic interactions show that Moody and PKA act antagonistically in BBB maintenance. They find that both too little and too much PKA activity disrupts the BBB. The authors also generate a serial section Transmission Electron Microscopy (TEM) volume to analyze wild type, PKA hyperactivity, and PKA loss of function animals. They find that loss of BBB function is due to gaps in the BBB, rather than thinning of the BBB. This conclusion is somewhat weak, however, because they did not analyze a genotype that has thin BBB structure but normal BBB function. Their work raises the question of how does PKA promote BBB integrity. They analyze two potential PKA targets (myosin light chain kinase [MLCK], and Rho1), finding that reduced Moody levels lead to disrupted subcellular localization of both proteins in SPG; that reducing MLCK or Rho1 levels causes failure of BBB function; and that reducing Moody can rescue these phenotypes. They conclude that Moody acts antagonistically to PKA/MLCK/Rho1 to establish distinct apical and basal membrane domains in SPG which are required for BBB function.

**Decision letter after peer review:**

Thank you for submitting your article "The cAMP effector PKA mediates Moody GPCR signaling in *Drosophila* blood-brain barrier formation and maturation" for consideration by *eLife*. Your article has been reviewed by 3 peer reviewers, including Chris Doe as the Reviewing Editor and Reviewer #1, and the evaluation has been overseen by Utpal Banerjee as the Senior Editor. The following individuals involved in review of your submission have agreed to reveal their identity: Marc Freeman (Reviewer #2); Sarah Ackerman (Reviewer #3).

Essential Revisions:

1) On page 10 the authors say "Thus, PKA signaling profoundly reorganizes the actin and MT cytoskielotn, thereby regulating the membrane overlap formed between neighboring SPG." Yet Figure 3 to me does not conclusively show membrane overlap. The figure shows surface views of the CNS, whereas overlap occurs on the Z-axis in this orientation. The figure shows nothing about the overlap, and the legend merely describes the markers without guidance on how to interpret the staining relative to overlap (if possible). Thus, I think relevant data should be newly provided on overlap, or the wording toned down in the Results section and the legend improved.

2) The study needs to be strengthened by more rigorous quantification. There is no quantification in figure 3. This is a primary point in the manuscript-that cytoskeletal markers change (in a claimed "monotonic" way) in SPGs when PKA is altered. There is also no quantification or statistics in Figure 5, which is among the most interesting observations. This is essential. It is also difficult to understand how to do this most reasonably and rigorously. The nucleus provides a spot to see a nice separation of apical and basal membranes, but is this asymmetric localization true across the entire BBB? This is challenging, but finding a way would make the claims more convincing.

While the authors note the percentage of animals that show BBB penetration defects (Figure 1B,G; Figure 2D,G; Figure 6A,E; Figure S1), it is not apparent whether the statistics were applied only to animals that exhibited BBB defects, or to the entire cohort of animals. Additionally, while the authors give a range of N values per experiment, what N signifies is not clearly defined (animals, septate junctions, etc). These comments could easily be addressed by text. Finally, it is now standard in the field to show individual data-points within bar graphs to show the spread of the data more precisely. Changing the format of the graphs throughout to display the raw data would improve our understanding of the comparisons made (re: first point above).

3) Given their beautiful supplemental live imaging experiments, Figure 3 could be greatly improved by live imaging of vesicle dynamics to see if the PKA-induced changes in localization are progressive, or impaired from the start.

4) In Figure 5, while it is clear that Moody and PKA are not co-localized, it is difficult to conclude that they are localized at the apical versus basal side of the cell without membrane markers. Additional analyses with membrane markers would greatly clarify this result.

5) As PKA signaling is also known to interact with the small GTPase Rac, and Rac is also known to modify cytoskeletal dynamics, Figure 6 would be improved by epistasis experiments with PKA and Rac mutants; if these are not readily available, the experiment could be mentioned in the Discussion as a future direction.

6) In the manuscript, the authors note that PKA has an unexpected role in dictating cell polarity, and in membrane protrusion. These features of PKA are also true for Schwann cells during development (reviewed in PMC5181106, PMC4526746). The authors should highlight these similarities, which would expand the scope and general readership for this article.

---

## [Author Response]

Essential Revisions:1) On page 10 the authors say "Thus, PKA signaling profoundly reorganizes the actin and MT cytoskielotn, thereby regulating the membrane overlap formed between neighboring SPG." Yet Figure 3 to me does not conclusively show membrane overlap. The figure shows surface views of the CNS, whereas overlap occurs on the Z-axis in this orientation. The figure shows nothing about the overlap, and the legend merely describes the markers without guidance on how to interpret the staining relative to overlap (if possible). Thus, I think relevant data should be newly provided on overlap, or the wording toned down in the Results section and the legend improved.

We agree that Figure 3 did not clearly show the membrane overlap which occurs along the Z-axis in this orientation, and we only saw a much wider and thicker belt in the contacting membrane area between neighboring SPG. However, we did show the extent of membrane overlap in response to different PKA activity levels by two methods: (1) The membrane marker GapGFP was introduced to show the plasma membrane overlap between neighboring SPG cells under different Moody/PKA activities by confocal microscopy in Figure 2C, but only with limited information due to the resolution of light microscopy; (2) And we further performed serial section TEM (ssTEM), followed by computer-aided reconstruction of TEM stacks to resolve the 3D ultrastructure of cell contacts and SJs under different PKA activity levels in third instar larva. These results, presented in Figure 4C-E, clearly show the membrane overlap between neighboring cells at the ultrastructural level (magenta and green shaded) upon different levels of PKA activity. In response to the reviewer’s comments, we changed the wording in both the result section and the legend accordingly.

2) The study needs to be strengthened by more rigorous quantification. There is no quantification in figure 3. This is a primary point in the manuscript-that cytoskeletal markers change (in a claimed "monotonic" way) in SPGs when PKA is altered.

We did quantification with all cytoskeletal markers used in figure 3, including GFPactin, TauGFP, EB1GFP, NodGFP, and two endosome markers Rab4RFP, Rab11GFP. Please see the result in the right column of Figure 3. Technical details are provided in the method section.

There is also no quantification or statistics in Figure 5, which is among the most interesting observations. This is essential. It is also difficult to understand how to do this most reasonably and rigorously. The nucleus provides a spot to see a nice separation of apical and basal membranes, but is this asymmetric localization true across the entire BBB? This is challenging, but finding a way would make the claims more convincing.

Regarding the question of the asymmetric localization across the BBB, we performed plot profiles across different areas of SPG and clearly show that the asymmetric localization across the entire SPG is formed between neighboring cells, not only around the nucleus. We further examined the subcellular distribution of Moody and Pka-C1 under gain- or loss-of-function of Moody/PKA signaling in SPG, including Moody knockdown (moody>MoodyRNAi), PkaC1 Knockdown (moody>Pka-C1-RNAi), GPCR gain-of-function (moody>Go-GTP), and PKA overexpression (moody>mPka-C1*). We provide lateral views of the CNS/hemolymph border, in each condition, and did line scans of fluorescence intensities for each channel along the apical-basal axis to visualize the subcellular distributions along the apical-basal axis of SPG and the results clearly demonstrated that this polarization of Moody and PKA depends on the activity of Moody/PKA signaling pathway, please see the detail in the figure 5.

While the authors note the percentage of animals that show BBB penetration defects (Figure 1B,G; Figure 2D,G; Figure 6A,E; Figure S1), it is not apparent whether the statistics were applied only to animals that exhibited BBB defects, or to the entire cohort of animals. Additionally, while the authors give a range of N values per experiment, what N signifies is not clearly defined (animals, septate junctions, etc). These comments could easily be addressed by text.

We added N values for each experiment in text and figure legends; please see the changes in Figure legends in Figure 1B and 1G; Figure 2D and 2G; Figure 6A and E; Figure1—figure supplement 1. The statistics in all these figures were applied the entire cohort of animals (including the animals without BBB penetration defects).

Finally, it is now standard in the field to show individual data-points within bar graphs to show the spread of the data more precisely. Changing the format of the graphs throughout to display the raw data would improve our understanding of the comparisons made (re: first point above).

We appreciate the suggestion and now changed the format of the graphs to show the individual values in each experiment, as the reviewer suggested. In the dye penetration assay, only a small number of animals did not show defects. Therefore, we list the percentage of animals that had no BBB permeability defects and average the mean intensity of the dye penetrated into the brain in the whole group (including the ones with normal BBB function).

3) Given their beautiful supplemental live imaging experiments, Figure 3 could be greatly improved by live imaging of vesicle dynamics to see if the PKA-induced changes in localization are progressive, or impaired from the start.

Thank for the suggestion! Unfortunately, we did not succeed in live imaging vesicle dynamics for this point. The live imaging experiments in the supplemental material was done on whole animals at the end of embryogenesis (stage 17). However, the vesicle maker used in Figure 3 was used in third instar larvae, driven by moody-gal4. This marker is not sufficiently expressed for live imaging the embryonic stages; this stage is too early to visualize all the markers used for the larval experiments in Figure 3. We also tried a stronger driver (Repo-Gal4) to express vesicle markers in the embryo (stage 17), including Rab4-RFP and Rab11-GRP. However, they formed aggregates and did not mimic in vivo protein distribution in SPG. As pointed out in the paper, the reason we did this analysis in third instar larvae is that SPG double in size and are accessible via dissection of the CNS, which greatly facilitates the microscopic analysis with all live markers. Considering this situation and to make this data more convincing, we did quantification of all the markers to demonstrate the changes of vesicle markers in response to different PKA levels; this information is provided in Figure 3E and 3F.

4) In Figure 5, while it is clear that Moody and PKA are not co-localized, it is difficult to conclude that they are localized at the apical versus basal side of the cell without membrane markers. Additional analyses with membrane markers would greatly clarify this result.

We previously demonstrated that Moody localizes at the apical side of SPG by co-labeling with membrane marker GapGFP and SJ marker NrgGFP (Schwabe et al., 2017)(Figure 5). This polarized distribution of Moody depends on SJ organization, and Moody is excluded from the lateral membrane compartment and the basal membrane. We added this information in the revised version. In addition, we further examined the subcellular distribution of Moody and Pka-C1 under gain- or loss-of-function conditions of Moody/PKA signaling in SPG. WE also performed plot profiles of lateral compartments around the nucleus of SPG to visualize the subcellular distribution along the apical-basal axis of SPG. Our results clearly demonstrate that this polarization of Moody and PKA depends on the activity of Moody/PKA signaling pathway, and we have revised Figure 5 accordingly.

5) As PKA signaling is also known to interact with the small GTPase Rac, and Rac is also known to modify cytoskeletal dynamics, Figure 6 would be improved by epistasis experiments with PKA and Rac mutants; if these are not readily available, the experiment could be mentioned in the Discussion as a future direction.

We agree with the reviewer and now performed experiments to investigate the function of Rac1 in BBB: (1) we knocked down Rac1 with RNAi (Repo>Rac1-RNAi) in SPG and did not observe any noticeable effect. On the other hand, knock-down of Rho1 in SPG was lethal, and the morphology of SPG changed when Rho1 was overexpressed; (2) We checked the expression of Rac1 in SPG (Moody>EGFP-Rac1) and found it to be located inside of SPG, but not in a specific pattern.

6) In the manuscript, the authors note that PKA has an unexpected role in dictating cell polarity, and in membrane protrusion. These features of PKA are also true for Schwann cells during development (reviewed in PMC5181106, PMC4526746). The authors should highlight these similarities, which would expand the scope and general readership for this article.

We appreciate the helpful suggestion. SJs are homologous structurally and molecularly to the vertebrate paranodal junction that forms between axons and myelinating glial cells（Schwann cells）adjacent to the node of Ranvier. We have now included this point in both the introduction and discussion.